



# Evaluating the quality of the E-OBS meteorological forcing data in EStreams for large-sample hydrology studies in Europe

Franziska Clerc-Schwarzenbach[*1], Thiago V. M. do Nascimento[*1,2]

[1]Department of Geography, University of Zurich, Zurich, 8057, Switzerland

[2]Eawag: Swiss Federal Institute of Aquatic Science and Technology, Dübendorf, 8600, Switzerland

[*]These authors contributed equally to this work.

*Correspondence to*: Thiago V. M. do Nascimento (thiago.nascimento@eawag.ch)

**Abstract.** To conduct large-sample hydrological studies over large spatial domains, standardized meteorological forcing data are often desired. For large-sample studies across Europe, the EStreams dataset and catalogue satisfies this demand. In ES-

treams, the meteorological time series are obtained from the Ensemble Observation (E-OBS) product which is available for all of Europe. Due to the large spatial extent of this dataset, limitations of data quality have to be expected when the dataset is compared to smaller-scale datasets, e.g., national level. In this study, we compare the meteorological time series included for 3423 catchments in EStreams to nine smaller datasets (mostly CAMELS datasets). We assess how the different meteorological data impact the performance of a bucket-type hydrological model. For most catchments, the precipitation amounts derived

from E-OBS are lower than the ones from the CAMELS datasets, while the temperature and the potential evapotranspiration values are higher. Model performances tend to be (slightly) lower when the E-OBS data are used than when the CAMELS datasets are used for calibration. Exceptions arise when the CAMELS data were derived from global datasets rather than national products, as well as when the station density in the E-OBS data is high. This study provides the first assessment of the E-OBS data at a continental scale for hydrological applications and shows that, despite some limitations, the dataset offers

a reasonable basis for large-sample hydrological modelling across Europe.

## 1 Introduction

Driven by their enormous value for hydrological modelling studies, large-sample hydrology (LSH) datasets have developed at a rapid pace in the past decades, and the development continues to gain momentum: Since 2017, more than a dozen "CAMELS" datasets were released or are being developed (Addor et al., 2017; Alvarez-Garreton et al., 2018; Bushra et al., 2025; Chagas

et al., 2020; Coxon et al., 2020a; Delaigue et al., 2025a; Fowler et al., 2021; Höge et al., 2023; Jimenez et al., 2025; Liu et al., 2025; Loritz et al., 2024; Mangukiya et al., 2025; Nijzink et al., 2025; Teutschbein, 2024a). Other animals entered the LSH stage too: Llamas (Helgason and Nijssen, 2024; Klingler et al., 2021a), a goat (*cabra* in Portuguese; Almagro et al., 2021), and a bull (Senent-Aparicio et al., 2024b).





In the past years, efforts also went into the creation of more overarching products, i.e., datasets covering not only one country
or region. The Caravan dataset (Kratzert et al., 2023) combined the streamflow data from thousands of catchments in already
published open source LSH datasets with standardized meteorological time series and catchment attributes from the global
ERA5-Land reanalysis (Muñoz-Sabater et al., 2021). Caravan is growing further and has become a quasi-global dataset (Färber
et al., 2024). For Europe, a dynamic standardized dataset and a catalogue that provides detailed guidance for retrieving stream-
flow data from national providers were introduced in EStreams (https://www.estreams.eawag.ch) by do Nascimento et al.
(2024).

Although these collections of large-sample datasets are valuable resources, the inclusion of an increasing number of catchments
in one dataset almost always goes hand in hand with difficulties in providing high-quality forcing data, due to the lower
availability of high-quality data for larger spatial extents. Furthermore, data processing choices (e.g., gap filling, interpolation)
are more often required at large scales and might introduce added uncertainty in the outcomes (McMillan et al., 2018).

In an earlier study, Clerc-Schwarzenbach et al. (2024) showed that the standardized meteorological data obtained from ERA5-
Land (Muñoz-Sabater et al., 2021) in the Caravan dataset (Kratzert et al., 2023) led to a consistently lower hydrological model
performance for catchments in the US, Brazil, and Great Britain, than when the meteorological forcing data from the corre-
sponding CAMELS datasets (Addor et al., 2017; Chagas et al., 2020; Coxon et al., 2020a) were used. This demonstrates the
importance of promoting awareness of the potential data quality losses when it comes to standardized meteorological data.

Similar to the standardization in Caravan, the meteorological data were also standardized in EStreams (do Nascimento et al.,
2024). For EStreams, the data were obtained from the European Ensemble Observation (E-OBS) product (Cornes et al., 2018).
After the publication of EStreams, questions on the quality of the meteorological forcing data from E-OBS arised in the LSH
community. Recent studies have evaluated the accuracy of the E-OBS precipitation product against reference datasets and
meteorological stations in some parts of Europe, including Greece (Mavromatis and Voulanas, 2021), the central Alps, eastern
Europe and Scandinavia (Bandhauer et al., 2022). These evaluations indicated that the quality of the E-OBS precipitation data,
when compared to data from high-resolution datasets focusing on a smaller area, is higher in regions with a high density of E-
OBS stations, such as in central Europe, while the reanalysis product ERA5 (Hersbach et al., 2020) partly outperformed E-
OBS in regions with a sparse station network (Bandhauer et al., 2022). Yet, evaluations of the E-OBS data for a larger extent,
and specifically for hydrological modelling, remain unexplored.

To be able to inform the users of EStreams (and of the E-OBS data in general) about the effects of the standardized meteoro-
logical data on hydrological applications, a comparison to the meteorological data contained in different national and regional
datasets (i.e., CAMELS datasets and similar products) is required.

For this study, we used 3423 catchments from nine European countries to assess the quality of the meteorological data provided
in EStreams (obtained from E-OBS). We did so by comparing the meteorological forcing data from E-OBS to the analogous
data contained for the same catchments in national or regional datasets, namely in LamaH-CE (Klingler et al., 2021a) for
Austria, CAMELS-DK (Liu et al., 2025) for Denmark, CAMELS-FR (Delaigue et al., 2025a) for France, CAMELS-DE (Loritz
et al., 2024) for Germany, CAMELS-GB (Coxon et al., 2020a) for Great Britain, the BULL Database (Senent-Aparicio et al.,



2024b) for Spain, CAMELS-SE (Teutschbein, 2024a) for Sweden, and CAMELS-CH (Höge et al., 2023) for Switzerland. In addition, we also included catchments from Czechia, with data from the not yet published CAMELS-CZ dataset (Jenicek et al., 2024).

Except for LamaH-CE (which includes globally available ERA5-Land data), the meteorological data in the smaller datasets stem from sources that were created specifically for the respective country. Following the approach of Clerc-Schwarzenbach et al. (2024), we did not only compare the meteorological data itself, but also the model performances that were achieved with the different meteorological forcings (but same streamflow data) when calibrating the bucket-type HBV (Hydrologiska Byråns Vattenbalansavdelning) model (Bergström, 1992, 1995; Seibert and Vis, 2012). This allowed us to assess the overall hydrological efficacy of the meteorological forcing data. We hypothesized that (i) the smaller datasets would provide a higher model performance than the standardized one; and that (ii) the difference in model performance introduced by the different forcing data would be smaller where E-OBS gauge densities are higher.

## 2    Data and Methods

### 2.1    Subset of catchments

We conducted this study for 3423 catchments that are available in the EStreams dataset and catalogue and in one of the following datasets: LamaH-CE, CAMELS-CZ, CAMELS-DK, CAMELS-FR, CAMELS-DE, CAMELS-GB, BULL, CAMELS-SE, CAMELS-CH, for simplicity's sake referred to as the "CAMELS datasets" throughout the remainder of the paper (Table 1). These catchments fulfilled the following cascade of criteria (with the number of catchments that were still included after each step in brackets):

- Located in a country with access to a CAMELS dataset at the time of data preparation, i.e., Austria, Czechia, Denmark, France, Germany, Great Britain, Iceland, Spain, Sweden, or Switzerland [12 019]
- High-quality catchment delineation in EStreams, as described by do Nascimento et al. (2024) [10 434]
- Catchment area (obtained from EStreams) below 2000 km$^2$ [9115]
- E-OBS meteorological time series available between October 1990 and September 2015 [8525]
- No redundancy among the catchments (the catchment with the longer streamflow time series was kept) [8367]
- Availability of at least 90 % of the $E_{pot}$ data between October 1990 and September 2015 [8297]
- Availability of the catchment in one of the CAMELS datasets [4246]
- Availability of at least 90 % of the streamflow data (in the CAMELS dataset) between October 1995 and September 2015 [3614]
- Average streamflow (in the CAMELS dataset) between October 1995 and September 2015 below 10 mm d$^{-1}$ [3608]
- Runoff ratio (in the CAMELS dataset) between October 1995 and September 2015 below 1.1 [3563]
- Runoff ratio (in EStreams) between October 1995 and September 2015 below 1.1 [3423]



Note that no catchments were excluded due to a high human influence. While identifying such influences for a large number
of catchments is possible, it remains a challenging task (Klotz et al., 2025; Senent-Aparicio et al., 2024b). Given these diffi-
culties and the fact that such influences will affect model performances with all types of forcing data, we chose not to apply
this filter.

**Table 1: Overview of catchments and data sources used in this study.**

| Country | Number of catchments included in this study | CAMELS dataset | Publication | Dataset |
|---|---|---|---|---|
| Austria | 428 | LamaH-CE | Klingler et al. (2021a) | Klingler et al. (2021b) |
| Czechia | 297 | CAMELS-CZ | *unpublished* | *unpublished* |
| Denmark | 139 | CAMELS-DK | Liu et al. (2025) | Koch et al. (2025) |
| France | 515 | CAMELS-FR | Delaigue et al. (2025a) | Delaigue et al. (2025b) |
| Germany | 1054 | CAMELS-DE | Loritz et al. (2024) | Dolich et al. (2024) |
| Great Britain | 560 | CAMELS-GB | Coxon et al. (2020a) | Coxon et al. (2020b) |
| Spain | 245 | BULL | Senent-Aparicio et al. (2024b) | Senent-Aparicio et al., (2024a) |
| Sweden | 23 | CAMELS-SE | Teutschbein (2024a) | Teutschbein (2024b) |
| Switzerland | 162 | CAMELS-CH | Höge et al. (2023) | Höge et al. (2025) |

## 2.2  Meteorological data

For the data comparison and the modelling experiments, we investigated and used daily precipitation, $E_{pot}$, and temperature
data from the EStreams dataset and from the different CAMELS datasets (Table 1). In EStreams, precipitation and temperature
data were obtained from the E-OBS ensemble mean product with a spatial resolution of 0.25° in both latitude and longitude
(do Nascimento et al., 2024). E-OBS provides a pan-European observational dataset of surface climate variables that is derived
by statistical interpolation of in-situ measurements, collected from national data providers (Cornes et al., 2018). Potential
evapotranspiration ($E_{pot}$) time series in EStreams were calculated with the Hargreaves formula (Hargreaves and Samani, 1982),
using the E-OBS temperature data and catchment elevation as input. EStreams is a ready-to-use product derived from E-OBS
and is likely to be increasingly used by the LSH community for studies on European catchments. Therefore, we used the
meteorological data of precipitation, $E_{pot}$, and temperature as provided in EStreams for the evaluation of the E-OBS meteoro-
logical data. Note that there is also a version of E-OBS at a resolution of 0.1° available, but not represented in EStreams.
Similarly, there are different $E_{pot}$ products available from E-OBS, but here, we used the $E_{pot}$ product provided in EStreams.
The CAMELS meteorological data are usually based on in-situ observations, but some datasets also incorporate satellite or
reanalysis information at several resolutions. When more than one option was available, we chose the data with the highest
spatial and (original) temporal resolution to represent the CAMELS data for this study (Table 2). While E-OBS was developed



specifically for Europe, one can still expect a lower data quality than for datasets created for a smaller region (e.g., national datasets) due to the lower spatial resolution and interpolation choices used to achieve the larger spatial extent of the dataset.

**Table 2: Overview of the data sources for the meteorological data (precipitation *P*, temperature *T*, and potential evapotranspiration $E_{\mathrm{pot}}$) for the different CAMELS datasets in this study.**

| Country | Variable(s) | Source / equation | Resolution | Reference(s) |
|---|---|---|---|---|
| Austria | $P$, $T$ | ERA5-Land | 9 km | Muñoz-Sabater et al. (2021) |
| | $E_{\mathrm{pot}}$ | Thornthwaite | * | Thornthwaite and Mather (1957) |
| Czechia | $P$, $T$ | *unpublished* data from Czech Hydromete-orological Institute (M. Jeníček / O. Ledvinka, pers. comm.) | 500 m | Štěpánek et al. (2011, 2013) |
| | $E_{\mathrm{pot}}$ | *unpublished* data based on Oudin equation (M. Jeníček / O. Ledvinka, pers. comm.) | * | Oudin et al. (2005) |
| Denmark | $P$ | Danish Meteorological Inst. | 10 km | Scharling (1999b) |
| | $T$ | Danish Meteorological Inst. | 20 km | Scharling (1999a) |
| | $E_{\mathrm{pot}}$ | Makkink | 40 km | van Kraalingen and Stol (1997) |
| France | $P$, $T$ | SAFRAN by Météo-France | 8 km | Quintana-Seguí et al. (2008); Vidal et al. (2010) |
| | $E_{\mathrm{pot}}$ | Oudin | * | Oudin et al. (2005) |
| Germany | $P$ | HYRAS by Deutscher Wetterdienst (DWD) | 1 km | Rauthe et al. (2013) |
| | $T$ | HYRAS by DWD | 5 km | Razafimaharo et al. (2020) |
| | $E_{\mathrm{pot}}$ | Modified Hargreaves | * | Adam et al. (2006); Droogers and Allen (2002); Hargreaves and Samani (1982) |
| Great Britain | $P$ | CEH-GEAR | 1 km | Keller et al. (2015); Tanguy et al. (2016) |
| | $T$ | CHESS-met | 1 km | Robinson et al. (2017a) |
| | $E_{\mathrm{pot}}$ | CHESS-PE (based on Penman-Monteith) | 1 km | Robinson et al. (2016, 2017b) |
| Spain | $P$, $T$, $E_{\mathrm{pot}}$ | Spanish Meteorological Agency (AEMET) | 0.05 ° | Peral García et al. (2017) |
| Sweden | $P$, $T$ | PTHBV database by Swedish Meteorologi-cal and Hydrological Institute (SMHI) | 4 km | SMHI (2025) |
| | $E_{\mathrm{pot}}$ | *unpublished* data based on Hamon equation (C. Teutschbein, pers. comm.) | * | Hamon (1963) |
| Switzerland | $P$ | RhiresD by MeteoSwiss | 2 km | MeteoSwiss (2021b) |
| | $T$ | TabsD by MeteoSwiss | 2 km | MeteoSwiss (2021a) |
| | $E_{\mathrm{pot}}$ | Penman–Monteith without interception cor-rection | * | Viviroli et al. (2007) |

* calculation for each catchment based on its meteorological data



## 2.3 Calculations of the differences in the CAMELS and E-OBS data

We compared the precipitation, $E_{pot}$, and temperature data from EStreams (i.e., the E-OBS data) to the corresponding data from the different CAMELS datasets (Table 2) for the twenty years between October 1995 and September 2015 to get an overview of the differences in the data. For precipitation and $E_{pot}$, we determined the relative difference in the mean annual sums. For temperature, we determined the mean absolute difference for the daily data. When comparing the two datasets, we used the E-OBS data obtained from EStreams as minuend and the analogous data obtained from the CAMELS datasets as subtrahend, i.e., positive differences indicate higher values in the E-OBS data, while negative differences indicate lower values in the E-OBS data than the CAMELS data. To calculate relative differences (for precipitation and $E_{pot}$), we divided by the mean annual sum determined from the CAMELS dataset. Thus, for example, a value of -20 % indicates that the mean annual sum obtained from E-OBS is 80 % of the mean annual sum obtained from the CAMELS dataset, and a value of 40 % indicates that the mean annual sum obtained from E-OBS equals 140 % of the mean annual sum obtained from the CAMELS dataset.

## 2.4 Modelling experiments

We defined different combinations of forcing data ("scenarios") to calibrate the hydrological model (Table 3). This allowed us to determine how the forcing data from E-OBS and from CAMELS individually impacted hydrological model performance. Note that due to the dependency of the E-OBS $E_{pot}$ data on the E-OBS temperature data, model calibration was influenced by the E-OBS temperature data even when replacing the temperature data from E-OBS with those from CAMELS (scenario V). Analogously to Clerc-Schwarzenbach et al. (2024), we calibrated the HBV model (Bergström, 1992, 1995) in the version HBV-light (Seibert and Vis, 2012) with a genetic algorithm (Seibert, 2000). Each catchment was divided into elevation zones of 200 m elevation difference, whereby an elevation zone had to account for at least 5 % of the catchment area. For the determination of the elevation zones, we used the shapefiles provided by EStreams, and the Copernicus DEM (European Space Agency and Airbus, 2022).

We used the five years from October 1990 to September 1995 as the warming-up period for the model, and the twenty years from October 1995 to September 2015 as the simulation period for which we optimized daily streamflow simulation in terms of the Kling-Gupta efficiency KGE (Gupta et al., 2009). One calibration consisted of a total of 3500 model runs. We conducted each calibration ten times to account for equifinality. We used equal weights on the ten simulated hydrographs to calculate an ensemble mean hydrograph. We determined the model performance (using again the KGE) for each catchment and each scenario by comparing this ensemble mean hydrograph to the observed hydrograph.

**Table 3: Combinations of forcing data ("scenarios") for the modelling experiments.**

| Scenario | Description | Precipitation | $E_{pot}$ | Temperature |
|----------|-------------|---------------|-----------|-------------|
| I | CAMELS | CAMELS | CAMELS | CAMELS |
| II | E-OBS (EStreams) | E-OBS (EStreams) | E-OBS (EStreams) | E-OBS (EStreams) |



| III | E-OBS with CAMELS precipitation | CAMELS | E-OBS (EStreams) | E-OBS (EStreams) |
| IV | E-OBS with CAMELS $E_{pot}$ | E-OBS (EStreams) | CAMELS | E-OBS (EStreams) |
| V | E-OBS with CAMELS temperature | E-OBS (EStreams) | E-OBS (EStreams) | CAMELS |

# 3    Results

## 3.1    Comparison of the meteorological data

The mean annual precipitation sums in the E-OBS data were lower than the mean annual precipitation sums in the CAMELS data for 3042 catchments (89 %). For 1506 catchments (44 %), the deviation of the mean annual precipitation sums in E-OBS from the ones in CAMELS exceeded -10 %. Conversely, there were only 57 catchments (2 %) for which the mean annual precipitation sums in E-OBS were overestimated by +10 % or more from the ones in CAMELS. Differences between the two data sources were largest for the catchments in Spain and smallest for the catchments in Germany (Fig. 1).

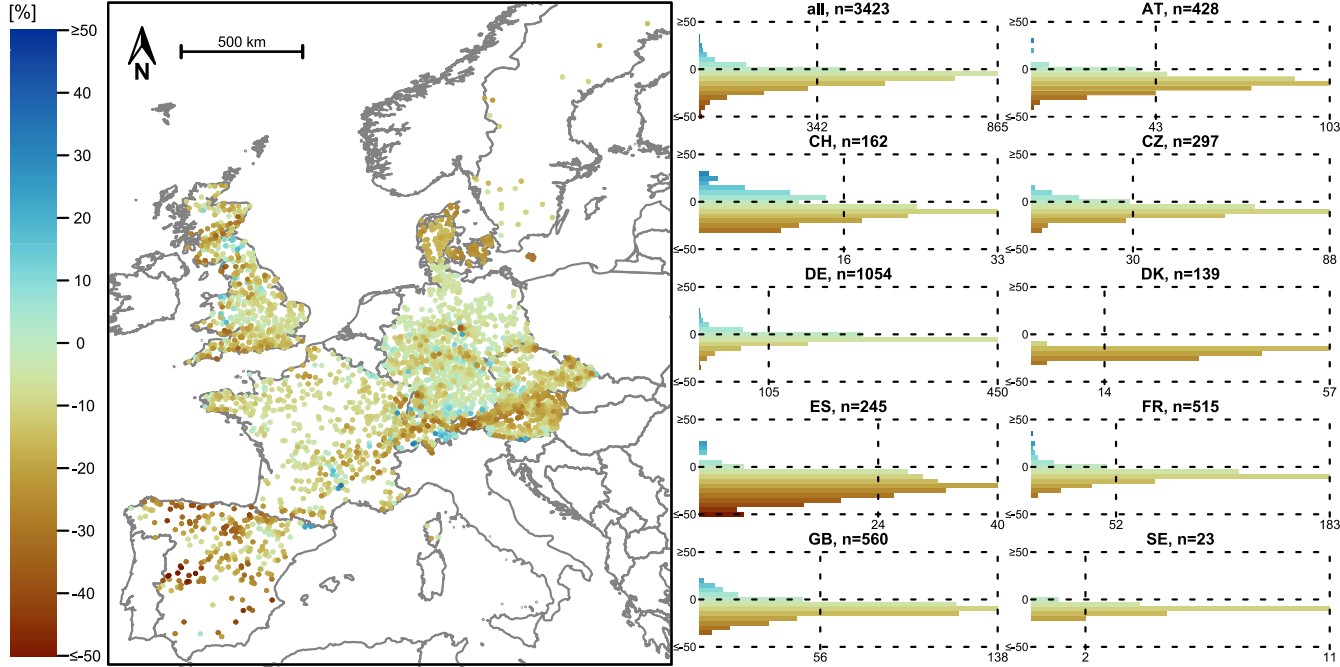

**Figure 1: Relative difference in the mean annual precipitation (for a 20-year period: 1995-2015) between the E-OBS data obtained from EStreams and the different CAMELS datasets. Negative values and brown colours indicate less precipitation in E-OBS than in CAMELS, positive values and blue colours more precipitation in E-OBS. On the map, the catchments with the largest deviations were plotted last to increase their visibility. Note that the number of catchments per histogram differ. This is illustrated by the vertical lines indicating 10 % (rounded) of the number of catchments per histogram. The colour scale was cut at ±50 %. The scale bar refers to the map centre. The base map was obtained from Natural Earth (naturalearthdata.com). The colour palette used in this and all other maps are scientific colour palettes from Crameri (2023).**



The opposite was found for the annual sums of $E_{pot}$: For 3215 catchments (94 %), the mean annual $E_{pot}$ calculated from the E-OBS data was higher than for the CAMELS data. For 2012 catchments (59 %), the deviation of the E-OBS $E_{pot}$ sums from the

CAMELS $E_{pot}$ sums were at least 10 %. Lower $E_{pot}$ sums derived from E-OBS than from CAMELS could only be observed for catchments in Sweden, on the Danish islands, in the canton of Ticino in Switzerland, and for some catchments in northern Spain (Fig. 2). As different equations or data sources were used in the different CAMELS datasets (see Table 2) to obtain the $E_{pot}$ data, the order of magnitude of the deviations changed abruptly along the national borders in some cases (e.g., along the border between Czechia and Germany or Austria). It is noteworthy that for $E_{pot}$, there tend to be small differences between the

two datasets for Spain (while this was not the case for precipitation).

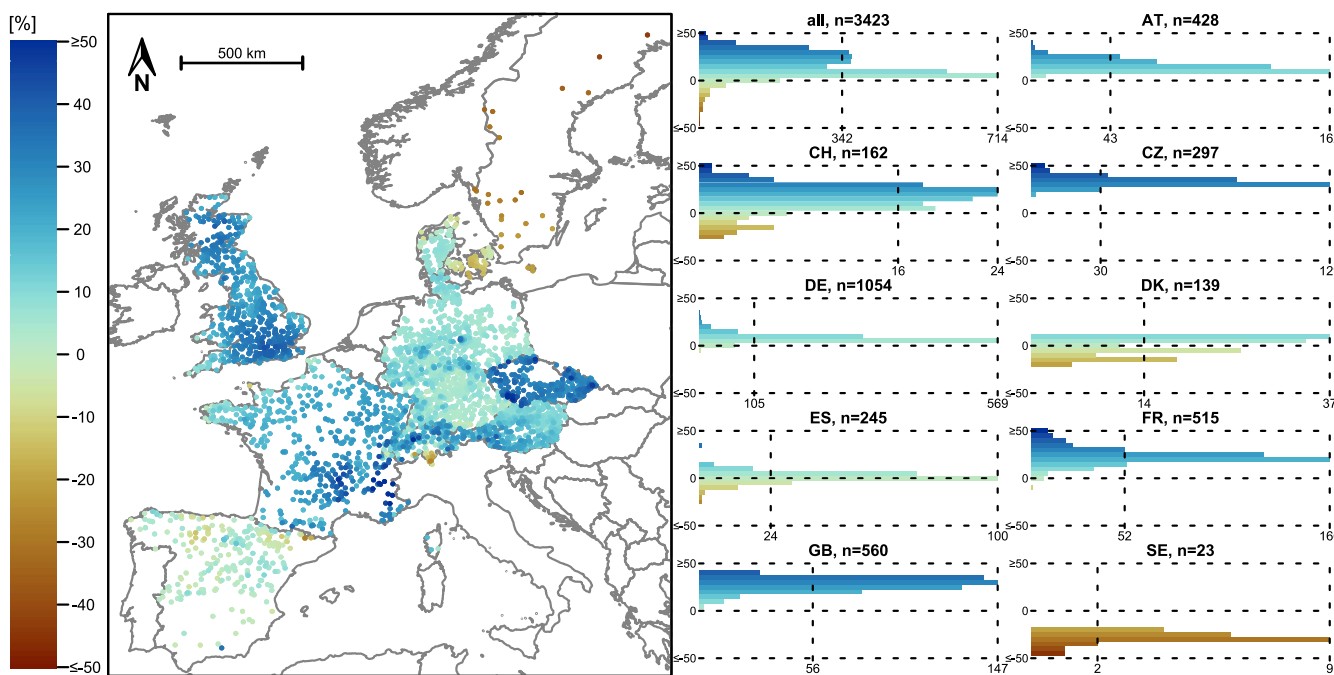

**Figure 2: Relative difference in the mean annual $E_{pot}$ (for a 20-year period: 1995-2015) obtained from EStreams and calculated from the E-OBS data compared to the mean annual $E_{pot}$ calculated from the different CAMELS datasets. Negative values and brown**

**colours indicate a lower $E_{pot}$ in E-OBS than in the CAMELS datasets, positive values and blue colours a higher $E_{pot}$. On the map, the catchments with the largest deviations were plotted last to increase their visibility.**

Due to the differences in the precipitation and the $E_{pot}$ data, the aridity indices ($E_{pot}/P$) calculated from the two data sources differed, although they were still highly correlated (Pearson's correlation coefficient of 0.93) (Fig. 3). Given the lower precip-

itation and higher $E_{pot}$ sums for most catchments, the aridity indices were generally higher when the E-OBS data obtained from EStreams were used than when the CAMELS data were used. This did not apply for Sweden, as the $E_{pot}$ sums based on E-OBS were lower than the ones from CAMELS for this country. The two calculated aridity indices aligned best for Germany and worst for Spain, Great Britain, and Czechia. Spatially, the aridity indices derived from both datasets followed the expected





pattern, with more arid catchments in southern Europe and north-eastern Germany and more humid catchments in the other
regions (see Fig. A1 for the CAMELS data and Fig. A2 for the E-OBS data).

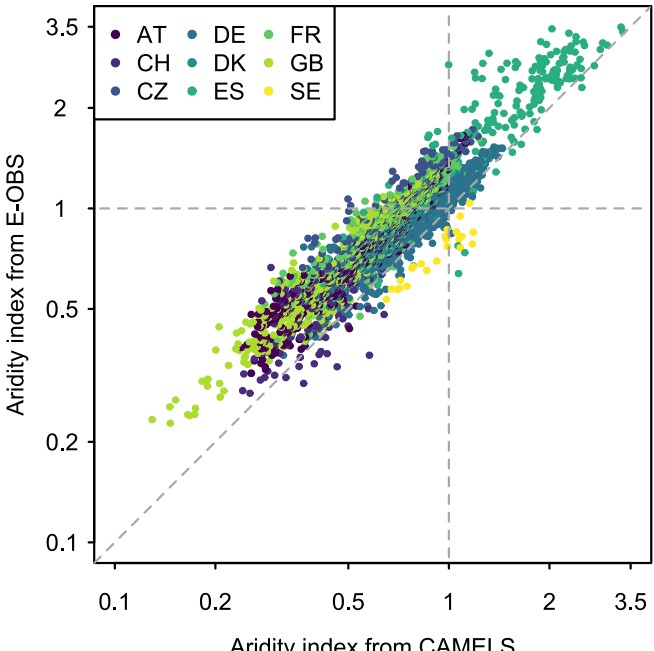

**Figure 3: Comparison of the aridity indices ($E_{pot}/P$) based on the CAMELS and the E-OBS data (for a 20-year period: 1995-2015), colour-coded by country. Note the logarithmic axes.**


Comparison of the temperature data in the two datasets revealed that the average temperature in E-OBS was higher (median difference: 0.3° C) for the vast majority of catchments than the average temperature in CAMELS (Fig. A3). There were 537 catchments (16 %) for which the average temperature was lower in E-OBS than in the CAMELS datasets. Note that in the HBV model, temperature has an effect on the snow routine, with higher temperatures resulting in a larger fraction of precipi-
tation falling as rain (and thus faster streamflow generation). However, as the threshold temperature for the differentiation between rain and snow is adapted during calibration, it is expected that the model can compensate comparably well for biased temperature time series. Thus, the main effect of the differences in temperature are actually the differences in $E_{pot}$ which are highly affected by the temperature data used as input to the calculations (Fig. 2Figure 2).

### 3.2   Model performances

**3.2.1   Model performances with the CAMELS and the E-OBS forcing data**

Overall, high model performances were achieved for most catchments when the CAMELS data (scenario I) were used for model calibration (Fig. 4). For 3066 of the 3423 catchments (90 %) the KGE was higher than 0.70, with a median performance





of 0.88. Among the nine countries, the median KGE was the highest for the catchments in Sweden (0.92, 100 % of the catchments had a KGE above 0.70), followed by Great Britain (median 0.91, 93 % above 0.70) and France (median 0.90, 94 %
above 0.70).

Conversely, the model performances were the lowest for catchments in Austria and Spain. The median KGE for Austria was 0.78; only 69 % of the catchments achieved a KGE above 0.70. The pattern was similar for Spain, with a median KGE of 0.83 and 78 % of the catchments having a KGE above 0.70. However, there were also other regions in which clusters of catchments with low model performances could be observed, such as the karstic area around London and some parts of south-eastern
Switzerland.

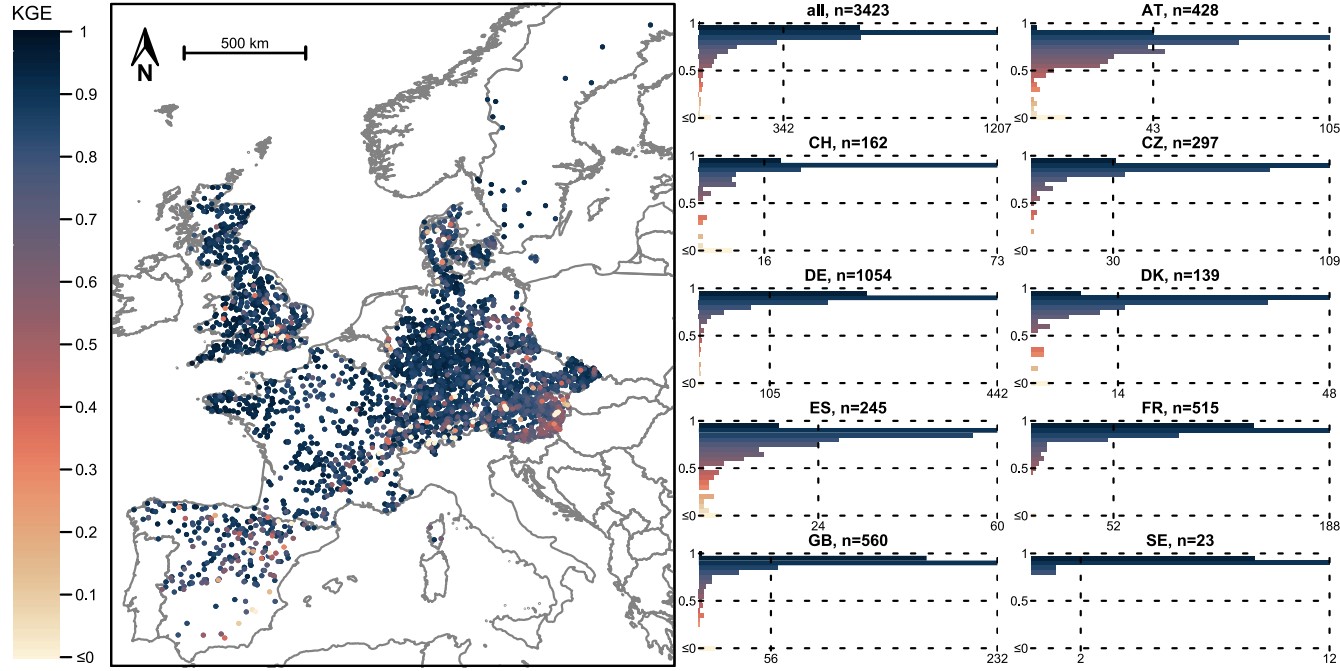

**Figure 4: Model performance (Kling-Gupta efficiency, KGE) achieved for the 20-year period between October 1995 and September 2015 when the input data from the CAMELS datasets were used for model calibration (scenario I). Note that the lower limit of the**
**colour scale was cut at zero. Lower performances were plotted last to improve their visibility.**

The model performances were also high for most catchments when the E-OBS forcing data (scenario II) were used for model calibration (Fig. 5). For 3059 of the catchments (90 %) the KGE was higher than 0.70, which is comparable to the 3066 catchments that fulfilled this criterion for the CAMELS data (scenario I). Furthermore, the median performance achieved with
the E-OBS data from EStreams (scenario II) of 0.87 was very similar to the one achieved with the CAMELS data (scenario I) (0.88). The median performance among all countries was still the highest for the catchments in Sweden (0.91).

Despite these overall similarities, there were also some notable differences in scenario II (E-OBS data) compared to scenario I (CAMELS data). The second-highest median KGE (0.91) was recorded for the catchments in France, where 97 % of the



catchments had a KGE above 0.70. After France, Germany followed with a median KGE of 0.88 and 95 % of the catchments
having a KGE above 0.70. For the catchments in Austria, where the precipitation data were obtained from ERA5-Land, clearly
higher model performances were achieved with the E-OBS data (scenario II) than with the CAMELS data (scenario I). The
median KGE in scenario II was 0.86 (scenario I: 0.78), and for 90 % of the Austrian catchments, the KGE was above 0.70
(scenario I: 69 %). However, for the catchments in Spain, the model performances were even lower than in scenario I: The
median KGE was 0.66 (scenario I: 0.83), and for only 40 % of the catchments, the KGE was above 0.70 (scenario I: 78 %).
Furthermore, for the catchments in the karstic area around London, the performances were higher with the E-OBS data (sce-
nario II) than with the CAMELS data (scenario I).

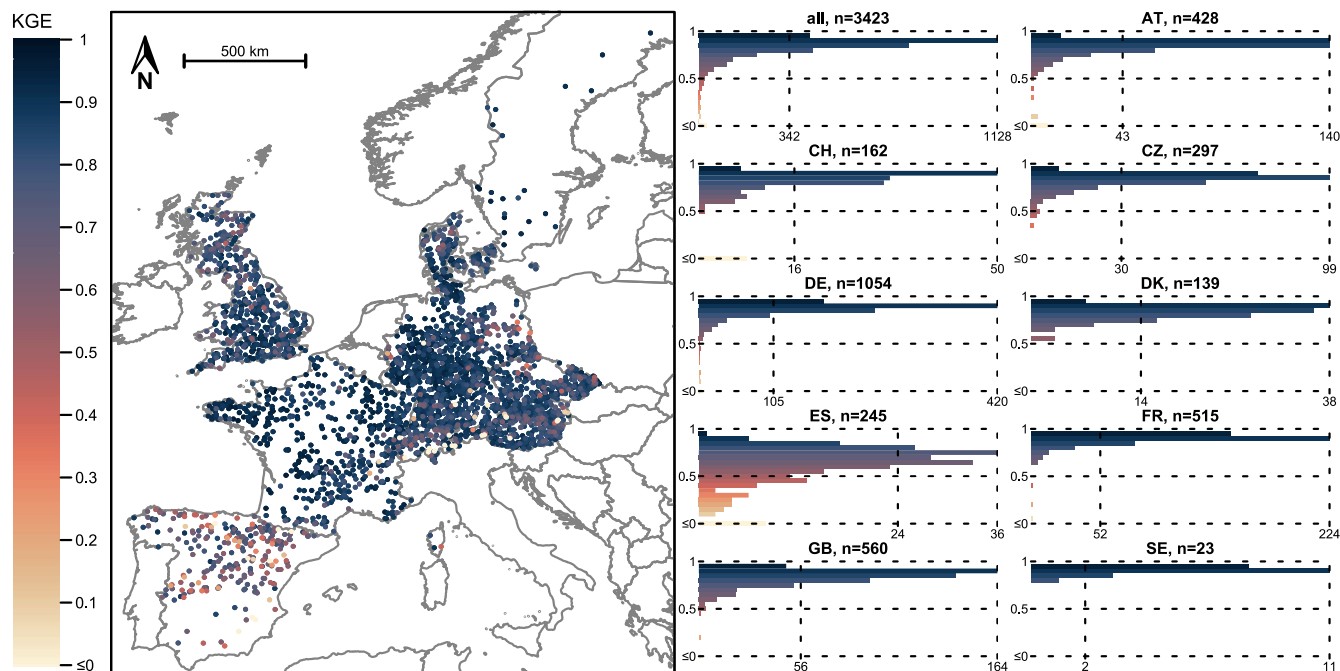

**Figure 5: Model performance (Kling-Gupta efficiency, KGE) achieved for the 20-year period between October 1995 and September**
**2015 when the E-OBS data obtained from EStreams were used for model calibration (scenario II). Note that the lower limit of the**
**colour scale was cut at zero. Lower performances were plotted last to improve their visibility.**

### 3.2.2    Differences in model performance between scenario II and I

To more easily assess the differences in model performances between scenario II and I, we looked at the differences in model
performance directly (Fig. 6). For 61 % of the catchments, model performances were (at least slightly) higher when the CAM-
ELS data were used (scenario I), while for the other 39 % of all catchments, the use of E-OBS data resulted in better model
performances. Not surprisingly, based on the results described above, the strongest regional differences were found for the



catchments in Austria, Spain, and Great Britain. When the catchments in Austria were excluded from the analysis (i.e., to only compare the E-OBS data with forcing data from smaller-scale datasets) the results were more pronounced: Then, 33 % of the catchments performed better with the E-OBS forcing data, while for 67 % of the catchments, model performances were higher with the CAMELS forcing data.

For two countries (Austria and France) there were notable improvements in model performance when using the E-OBS dataset. In Austria, 353 catchments (82 %) performed better with E-OBS forcing data obtained from EStreams ($\Delta$KGE>0), and the negative differences were not more pronounced than -0.15 (i.e., for the catchments with better performances in scenario I than in scenario II, the benefit was limited). In France, 331 catchments (62%) had higher performances with the E-OBS data, but the differences in performance were variable. For Sweden, 16 catchments (72 %) performed better with the E-OBS data, but the differences were very small (Fig. 6). For the catchments in Spain, it was the opposite: 220 catchments (92 %) performed better with the CAMELS dataset, and for some catchments, the differences were large. Similar trends were observed for the catchments in Great Britain (74 %), Germany (72 %), Switzerland (67 %), Czechia (62 %), and Denmark (57 %).

The results also indicated some interesting intracountry patterns (Fig. 6). In France, the most striking positive differences occurred for the catchments in the eastern part of the country, while for Great Britain, the CAMELS data resulted in clearly higher performances in most regions but not in the area around London. For the catchments in the center of Austria, the CAMELS data sometimes led to better model performances than the E-OBS data, while the opposite was the case in most other catchments (see above).



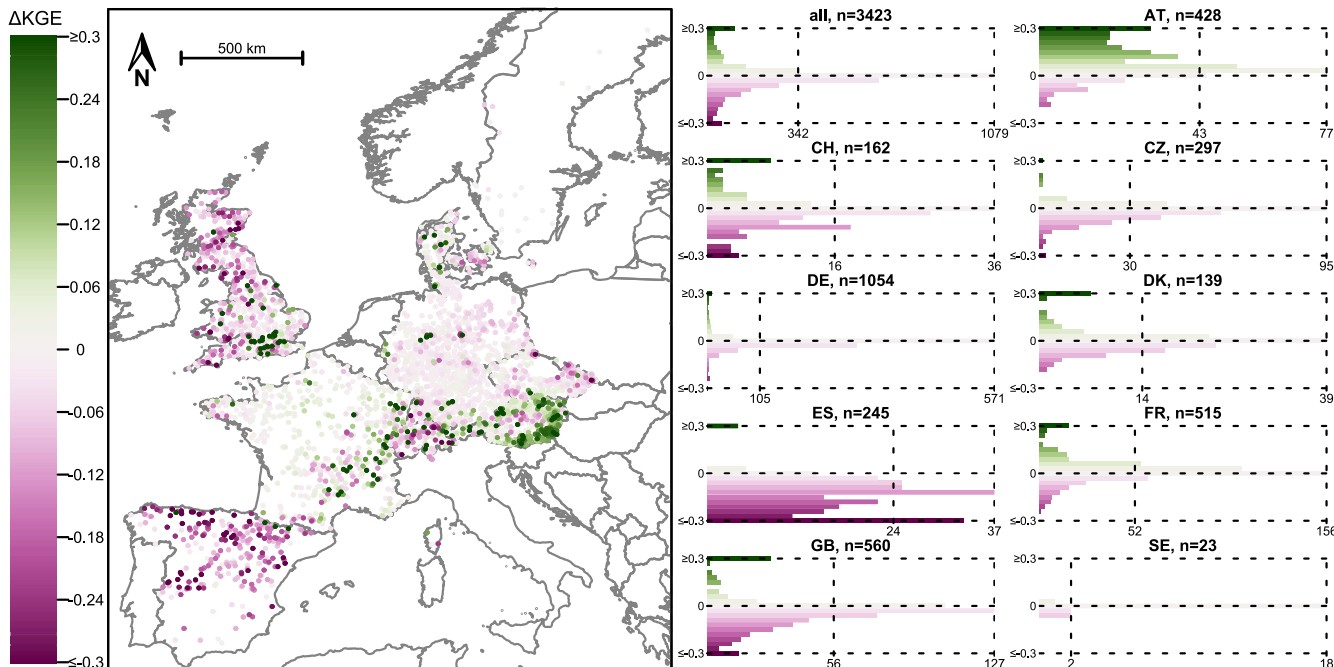

**Figure 6: Difference in model performance (Kling-Gupta efficiency, KGE) between scenario II and scenario I. Positive values and green colours indicate higher performances when the E-OBS data obtained from EStreams were used, negative values and pink colours indicate higher performances when the CAMELS data were used. For the model performances, see Figs. 4 and 5. The catchments with the largest differences (in absolute terms) were plotted last to improve their visibility. Note that the colour scale was cut at ±0.3.**

### 3.2.3 Differences in model performance between scenario II and scenarios III, IV, and V

Replacing precipitation from E-OBS with data from CAMELS had by far the strongest impact on model performance (scenario III, Fig. A4). For most catchments, the performance differences between scenarios II and III closely mirrored the performance differences between scenarios II and I, indicating that precipitation accounted for a large share of the overall differences in performance. For only a few catchments (mostly in Great Britain), was the performance gap between scenarios II and I notably larger than between scenarios II and III. The opposite occurred for very few catchments (Fig. A5).

The effect of replacing the $E_{pot}$ data was quite limited (scenario IV). The higher $E_{pot}$ data based on E-OBS were beneficial for a handful of catchments (ΔKGE>0.30 for 21 catchments), but the median difference was 0.00 (Fig. A6). Replacing only the temperature time series with the CAMELS data (scenario V) had virtually no effect on model performance for most catchments. There were no catchments for which the replacement of the temperature data increased the KGE by more than 0.10 and only five catchments for which it decreased the KGE by more than -0.10 (Fig. A7). Note that only the temperature time series provided as input data to the HBV model were replaced, and not the data that were used to calculate $E_{pot}$.



## 3.3 Model performance linked to catchment attributes

### 3.3.1 Number of E-OBS precipitation stations

It is well documented that the coverage of precipitation stations used to produce the gridded E-OBS dataset is highly variable (Bandhauer et al., 2022; do Nascimento et al., 2024). To assess the impact of this variability, we examined the relationships between the number of E-OBS stations within or near each catchment and the model performance for scenario II (i.e., using the E-OBS data contained in EStreams for all meteorological variables; see Fig. 5). Here, we present these relationship assessments per country (Fig. 7). The number of E-OBS precipitation stations was obtained from the EStreams dataset, defined as the count of stations located within a 10 km buffer of the catchment boundary (do Nascimento et al., 2024).

Model performances for scenario II tended to be higher when more E-OBS stations were located in or around a catchment. The performances were generally the lowest for areas with sparse station coverage, such as Spain and Great Britain. However, low E-OBS station density did not always result in poor model performance. For catchments in countries like France, Great Britain, Denmark, and Sweden, KGE values remained mostly above 0.5 despite a comparatively lower station density, suggesting that factors other than station density (such as the spatial variability of the rainfall due to topography or convective rainfall) also influence model accuracy. For further insights, we also provide the scatterplots of the differences in model performance between scenarios II and I compared to the number of E-OBS stations per catchment (Fig. A8).

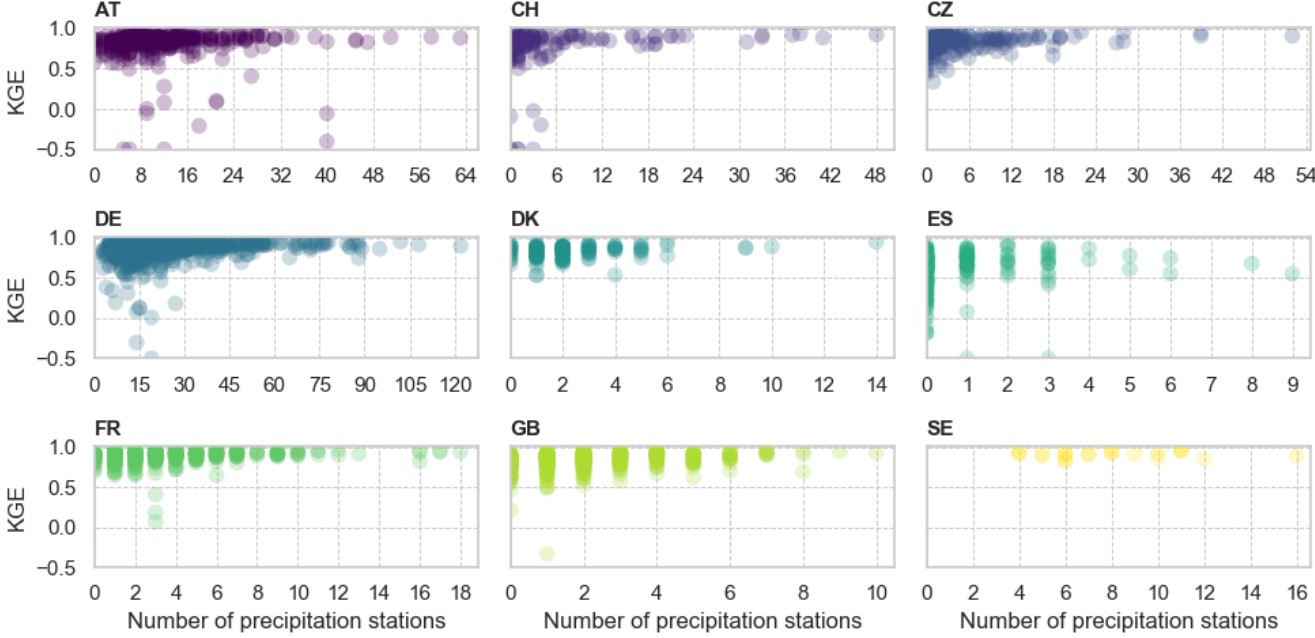

**Figure 7: Scatterplots showing the model performance (Kling-Gupta efficiency, KGE) for scenario II (y-axes) versus the number of precipitation stations used to derive the E-OBS precipitation data, per country. Each circle represents one catchment. Note that the y-axes were cut at -0.5, in accordance to Fig. 5, and the x-axes differ for the different subplots.**






### 3.3.2 Aridity index

We also evaluated the model performances for scenarios I and II in relation to the aridity indices derived from the respective forcing data (Fig. 8). Despite some atypical cases, the model performances tended to be lower in catchments with higher aridity indices (drier catchments). This trend was particularly evident for the catchments in Austria, Czechia, Germany, Spain, and

Sweden. Although the pattern appeared with both forcing datasets, it was more pronounced for the CAMELS data (scenario I), especially for the catchments in Austria, Czechia, and Spain.

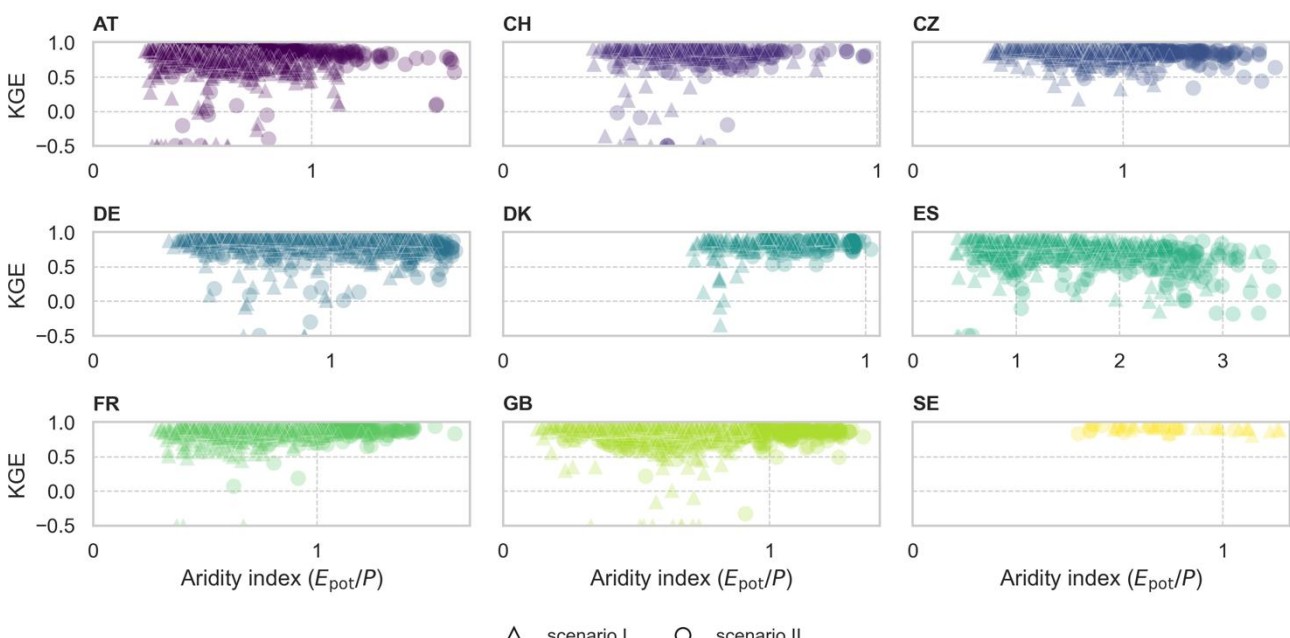

**Figure 8: Scatterplots showing the model performance (Kling-Gupta efficiency, KGE) for scenario I and II (y-axes) versus the aridity**
**indices derived from the corresponding forcing data (CAMELS for scenario I, E-OBS from EStreams for scenario II), per country.**
**Note that the y-axes were cut at -0.5, in accordance to Fig. 5. Note that the x-axes differ for the different subplots.**

## 4   Discussion

### 4.1   Model performances

The HBV model is known for its capabilities for simulating streamflow, particularly in humid catchments, where water flow is related to varying soil saturation and hydrological connectivity (Knapp et al., 2022, 2024). This, in part, seems to explain the consistently high model performances achieved using either the CAMELS or the E-OBS data (Figs. 4 and 5) for the wet





catchments, such as those in Sweden and Denmark. In contrast, in Spain, which has the most arid catchments, the KGE values were the lowest and most variable for both data scenarios, except for Austria under scenario I, where performance was also notably low. These findings are reinforced by the observed relationship between model performance and the aridity index, as shown in Fig. 8. The trend of decreasing performance with increasing aridity further supports the assertion that arid catchments pose significant challenges for hydrological modelling. Several other studies have suggested that for dry catchments more complex model structures may be needed for streamflow simulation, and even then, they still tend to yield lower model performance (Atkinson et al., 2002; David et al., 2022; Massmann, 2020).

Yet, the lower model performance for the catchments in Spain may be attributed not only to the inherent complexities of streamflow generation in arid environments, but also to the higher variability and limited availability of observational hydro-meteorological data in these regions, which complicates model calibration and validation, as noted in previous studies (do Nascimento et al., 2023; Yu et al., 2011). Additionally, previous studies have pointed out that many Spanish catchments, including the ones available in the currently used BULL dataset, are highly regulated, with dams and diversions. These heavily modified catchments may not be adequately filtered out in studies focused on natural hydrological simulation, thereby further impairing overall model performance (Klotz et al., 2025; Senent-Aparicio et al., 2024b).

## 4.2 Evaluation of the E-OBS data in comparison to the E-OBS station density

Our findings indicate that the model performance is strongly influenced by the density of stations used to obtain the E-OBS data. As a result, the reliability of model outputs varies considerably across regions—an observation that is consistent with previous research (Bandhauer et al., 2022; Klotz et al., 2025). This spatial dependency is visually supported by Figure 6 in the EStreams paper by do Nascimento et al. (2024), which shows the density of E-OBS stations across Europe. Notably, for regions with a high density of stations, such as Germany and Austria, the model reached the highest KGE values with E-OBS data, underscoring the critical role of data availability and quality in hydrological modelling accuracy.

As mentioned earlier, the results for the catchments in Austria are an interesting case, because the simulations with the E-OBS data (scenario II) were clearly better than those with the CAMELS data (scenario I). Unlike other CAMELS datasets, which rely on national meteorological data products, the CAMELS dataset for Austria (LamaH-CE) uses ERA5-Land as its meteorological forcing. Previous studies have already highlighted several limitations of ERA5-Land in simulating streamflow (Clerc-Schwarzenbach et al., 2024).

The fundamental differences between E-OBS and ERA5-Land stem from the data assimilation. The gridded product of E-OBS is based on a spatial interpolation of ground-based station networks, while ERA5-Land is a reanalysis dataset and derives precipitation from a model without direct incorporation of gauge data (Muñoz-Sabater et al., 2021). In densely monitored regions like Austria, this difference becomes especially relevant. This fundamental discrepancy, alongside the high E-OBS station density in Austria, seems to explain the superior performance with E-OBS data for the Austrian catchments.





### 4.3 Effect of different $E_{pot}$ data

In this work, the differences in model performance across the various sources of forcing data were attributed to the differences in precipitation inputs (scenario III), whereas discrepancies in $E_{pot}$ and temperature data hardly affected model performances (scenarios IV and V). These findings are consistent with those of Clerc-Schwarzenbach et al. (2024), who compared the Caravan and CAMELS datasets. Although their analysis reveiled larger differences in $E_{pot}$ than in precipitation data, it was still the precipitation inputs that exerted the greatest influence on model performance.

In this study, the differences between the $E_{pot}$ data derived from E-OBS and the $E_{pot}$ data derived from the CAMELS datasets were much smaller, but still striking (which makes sense, considering the differences in the temperature data that were generally used as input to the $E_{pot}$ calculations). This demonstrates once again the large uncertainties that we face when using different approaches to estimate $E_{pot}$. Several studies have already identified this issue as a persistent "blind spot" in hydrological modelling (e.g., Bai et al., 2016; Federer et al., 1996; Hanselmann et al., 2024). The $E_{pot}$ calculations for each catchment

in EStreams with the Hargreaves equation (do Nascimento et al., 2024) thus also affected the resulting $E_{pot}$ data that we used to represent the E-OBS $E_{pot}$. However, the Hargreaves equation was found to be reliable, among other regions especially in Central Europe (Pimentel et al., 2023) and this choice can therefore be supported. Furthermore, the differences in $E_{pot}$ data did not affect model performance results strongly, so it can be expected that the use of a different equation would not change the findings of this study.

While the different $E_{pot}$ data had a very limited effect on model performances in general, the higher $E_{pot}$ data derived from E-OBS were beneficial for the model performance in some cases, as they allowed the model more flexibility to adjust the water balance: For example, the low performances with the CAMELS-GB data for the catchments in the karstic area around London can (partially) be explained by the inability of the HBV model to simulate groundwater losses (Lane et al., 2019; Oldham et al., 2023; Seibert et al., 2018). In such catchments, the higher $E_{pot}$ values from E-OBS effectively helped to improve the water

balance by allowing for more evaporation, thereby compensating for the unmodelled groundwater losses. However, it is important to note that while this adjustment led to improved model performance, such compensatory effects are not desirable when the objective is to accurately represent internal catchment processes. Achieving realistic process representation should, generally, remain a central goal in hydrological modelling (Kirchner, 2006).

### 4.4 Limitations

As mentioned with the example of the compensation effects due to higher $E_{pot}$ data, a higher model performance does not necessarily mean a better representation of the hydrological processes. Still, we used the performance as an indicator for the hydrological efficacy of different forcing data. Apart the study by Clerc-Schwarzenbach et al. (2024), on which the methodology of this study was based, model performances are often used as an aggregated measure of data quality (Beck et al., 2017; Tarek et al., 2020). However, model performance can also be heavily influenced by the chosen model structure, particularly if





it does not align well with the physical characteristics of a given catchment. This structural sensitivity means that performance differences may reflect model limitations as much as data quality (Beven, 2018).

Another limitation is that we did not exclude any catchments due to human influences. Potentially, this affected model performances. However, this is the case then independent from the forcing data used, so we assumed that our conclusions would not change if these catchments were excluded.

Beyond model structure, it is worth noting that here we performed single-basin calibrations. While this allows for localized optimization, it does not reflect how models are typically regionalized for prediction in ungauged basins. Future research should explore how the identified performance patterns translate to a regionalization framework, which would provide more practical insights for prediction in data-scarce environments, and therefore, where model calibration is not possible.

In this study, we evaluated the E-OBS data at a resolution of 0.25°, as these are the data on which EStreams is based on (from

where we obtained the data for this study). Being included in EStreams, it can be expected that this E-OBS product will increasingly be used for LSH studies in Europe in the future. However, the E-OBS product with a higher resolution of 0.1° may lead to different results. Testing this is out of the scope of this study.

Finally, all simulations in this study were conducted at a daily time step. For smaller catchments, a finer temporal resolution, such as hourly, could provide more meaningful insights. With the increasing availability of temporally high-resolution datasets

(Coxon et al., 2025; Dolich et al., 2025; Nijzink et al., 2025), future studies may benefit from repeating similar analyses at the sub-daily timescale.

## 5    Conclusions

In this study, we compared the meteorological time series for 3423 European catchments in the EStreams dataset with time series from nine smaller-scale datasets (mostly CAMELS datasets). Moreover, we evaluated how the different meteorological

forcing data influence the performance of a bucket-type hydrological model.

Our results showed that, for the majority of catchments (89 %), annual precipitation values obtained from the E-OBS dataset were lower than those from the corresponding CAMELS datasets. The opposite was true for the annual sums of $E_{pot}$ and the average temperature (higher values in E-OBS than in CAMELS). These discrepancies led to consistently higher aridity indices computed with the E-OBS data in comparison to the CAMELS data for most catchments, although the spatial pattern remained

similar. Such systematic differences highlight important inconsistencies across the two data sources that can affect the outcomes of hydrological synthesis studies across large areas.

Despite these differences, model calibration using either set of forcing data achieved model performances above a KGE of 0.70 in more than 87 % of the catchments. However, performances were generally slightly lower when using E-OBS data than when using CAMELS data: For approximately 60 % of catchments, the model performance was higher when using CAMELS

forcing data. Notably, exceptions occured in cases where the CAMELS data were derived from global rather than national

products (ERA5-Land dataset in Austria). This aligns with our first hypothesis that smaller-scale, nationally curated datasets would lead to higher model performances than the standardized dataset.

Furthermore, our findings confirmed our second hypothesis that model performances using E-OBS forcing data would be lower in regions with a lower E-OBS station density. This highlights the critical role of station coverage on hydrological model
performance, an issue that becomes even more pronounced in mountainous regions, where steep climatic gradients in precipitation and temperature make dense and spatially representative data essential for reliable simulations, as well as in regions where convective rainfall is relevant.

Overall, this study presents the first assessment of the meteorological data from E-OBS across a continental domain and offers valuable insights for future large-sample hydrology studies.

**6    Data availability**

Except for CAMELS-CZ and the $E_{pot}$ data for CAMELS-SE, all the CAMELS data used are available from their respective repositories (see Table 1). The $E_{pot}$ data for Sweden were provided by Claudia Teutschbein and the CAMELS-CZ data by Michal Jeníček and Ondřej Ledvinka. These unpublished data are available upon reasonable request. The current version 1.2 of the EStreams dataset is available at a Zenodo repository (https://doi.org/10.5281/zenodo.14778580). The files containing
the model performances used in this study are stored at a GitHub repository (https://github.com/thiagovmdon/EOBS-quality).

**7    Code availability**

All code used to analyse the resolute and derive the figures for this manuscript are available at a GitHub repository (https://github.com/thiagovmdon/EOBS-quality).

**8    Author contribution**

TN designed the research question, TN and FCS prepared the data, FCS the model runs, TN and FCS evaluated the results and plotted them, TN and FCS wrote the manuscript.

**9    Competing interests**

The authors declare that they have no conflict of interest.





## 10 Acknowledgements

We acknowledge all teams that made the CAMELS datasets and the E-OBS data available. We acknowledge the co-authors with whom TN had developed the EStreams dataset. We thank the Science IT team of the University of Zurich for providing the infrastructure for cloud computing. We thank Ross Woods for awakening our curiosity about the research questions for this manuscript in the first place. We thank Marc Vis for providing us with data ready to be used in HBV for some CAMELS datasets. We thank Claudia Teutschbein for making the $E_{pot}$ data for CAMELS-SE available and Michal Jeníček and Ondřej

Ledvinka for providing us with the CAMELS-CZ data. We thank our supervisors Fabrizio Fenicia, Jan Seibert, and Ilja van Meerveld for fruitful discussions and their support, and Ilja van Meerveld also for her comments that helped to clarify the manuscript. TN was funded by a "Money Follows Cooperation" project (Project No. OCENW.M.21.230) between the Netherlands Organization for Scientific Research (NWO) and the Swiss National Science Foundation (SNSF).

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



## 12    Appendix

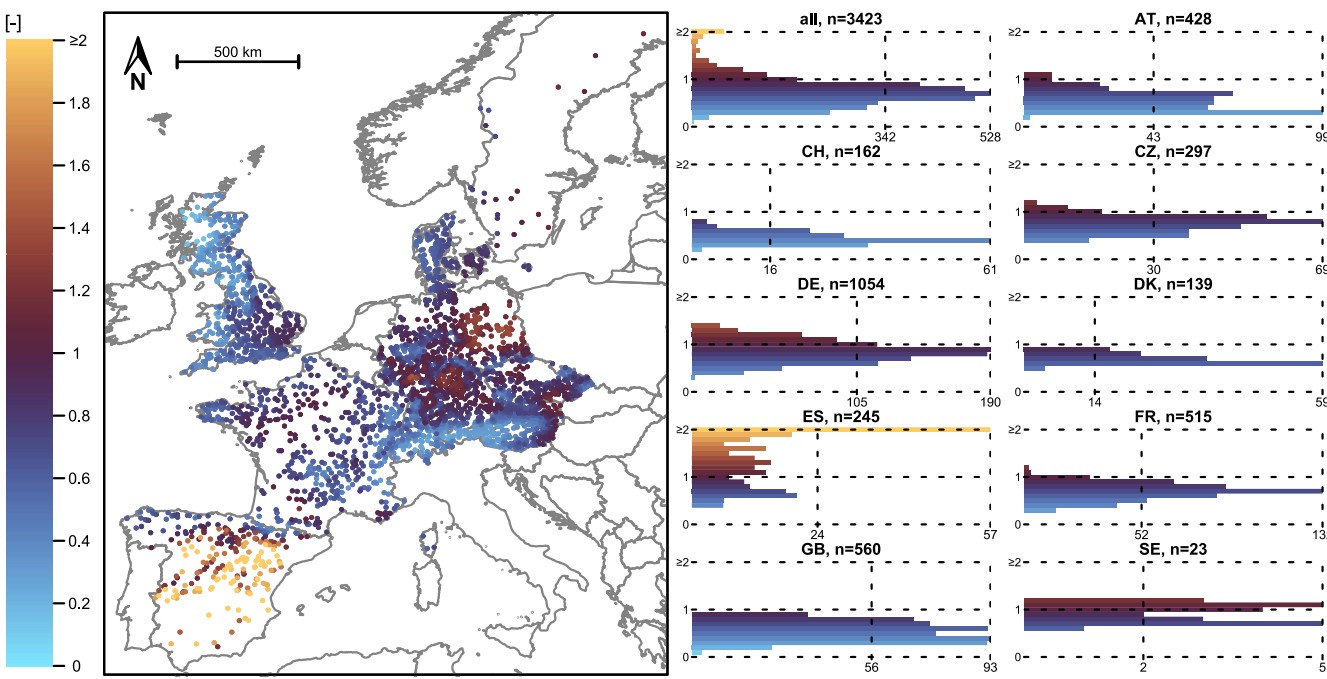

**Figure A1: Aridity index ($E_{pot}/P$) calculated from the CAMELS data (for a 20-year period: 1995-2015). Note that the colour scale**
**was cut at two.**



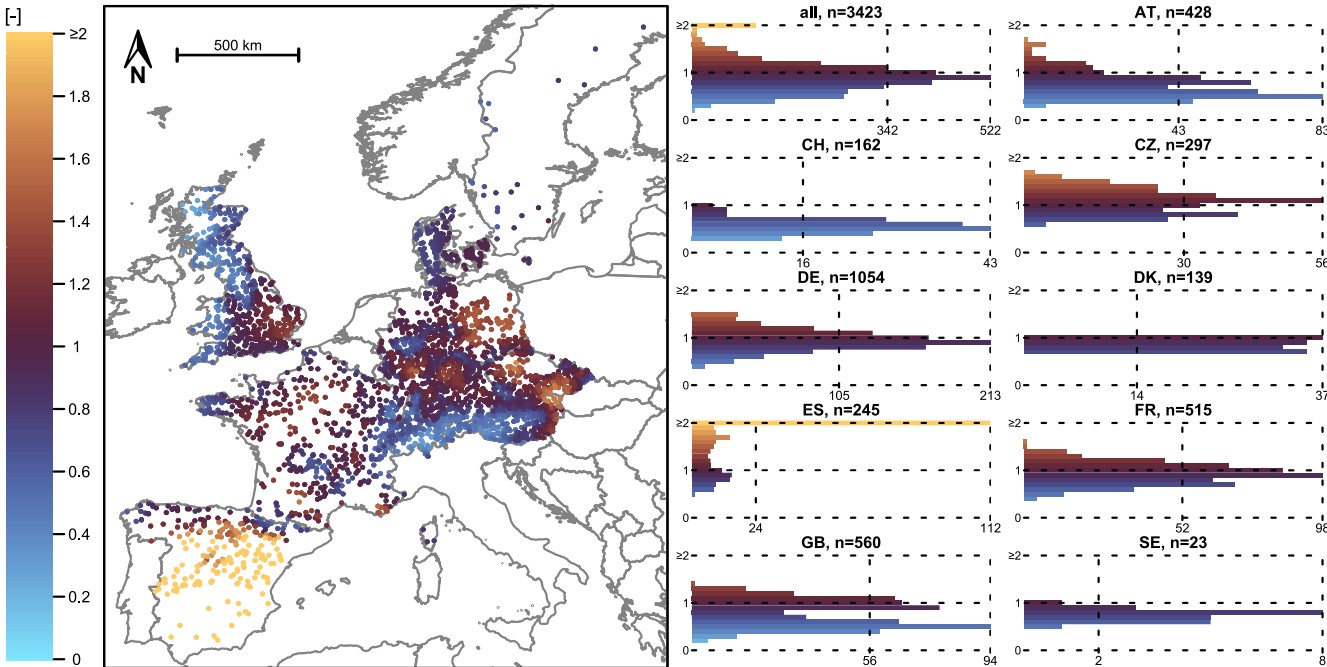

**Figure A2: Aridity index ($E_{pot}/P$) calculated from the E-OBS data obtained from EStreams (for a 20-year period: 1995-2015). Note that the colour scale was cut at two.**




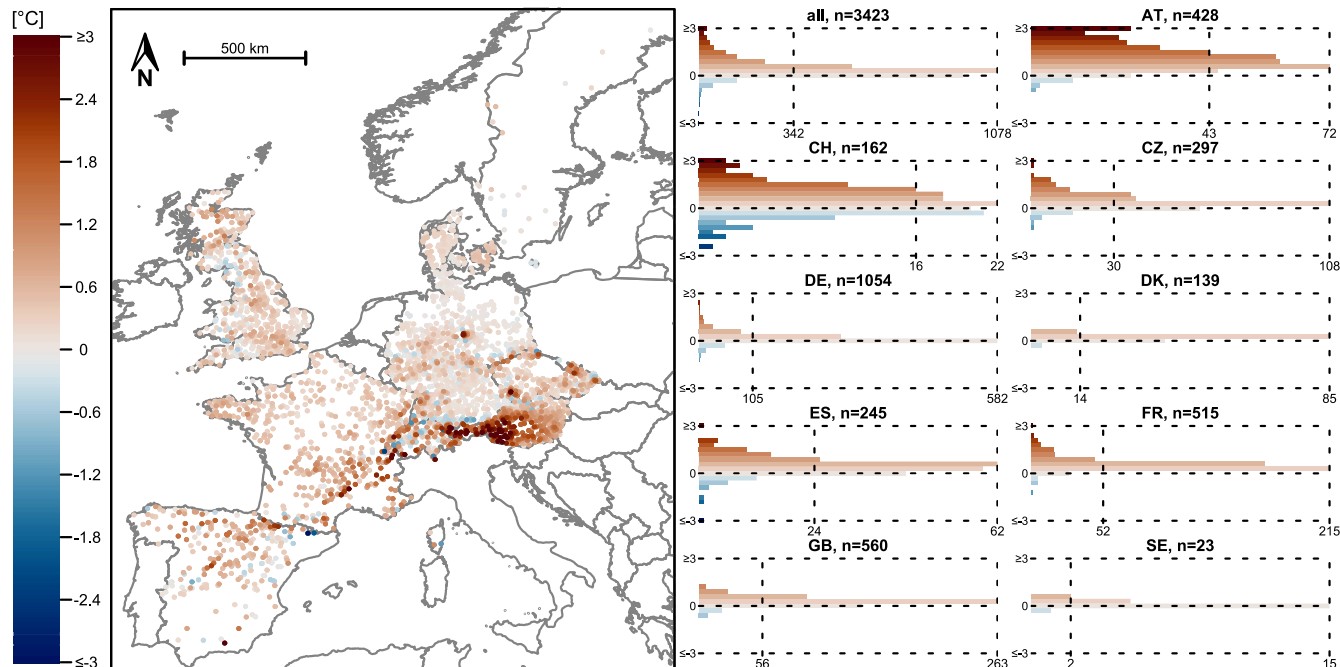

**Figure A3: Difference in mean annual temperature for each catchment when calculated from the E-OBS and the CAMELS datasets for the 20-year period 1995-2015. Positive values and red colours indicate higher temperatures in the E-OBS data obtained from EStreams, negative values and blue colours indicate lower temperatures in the E-OBS data. Note that the colour scale was cut at ±3 °C.**






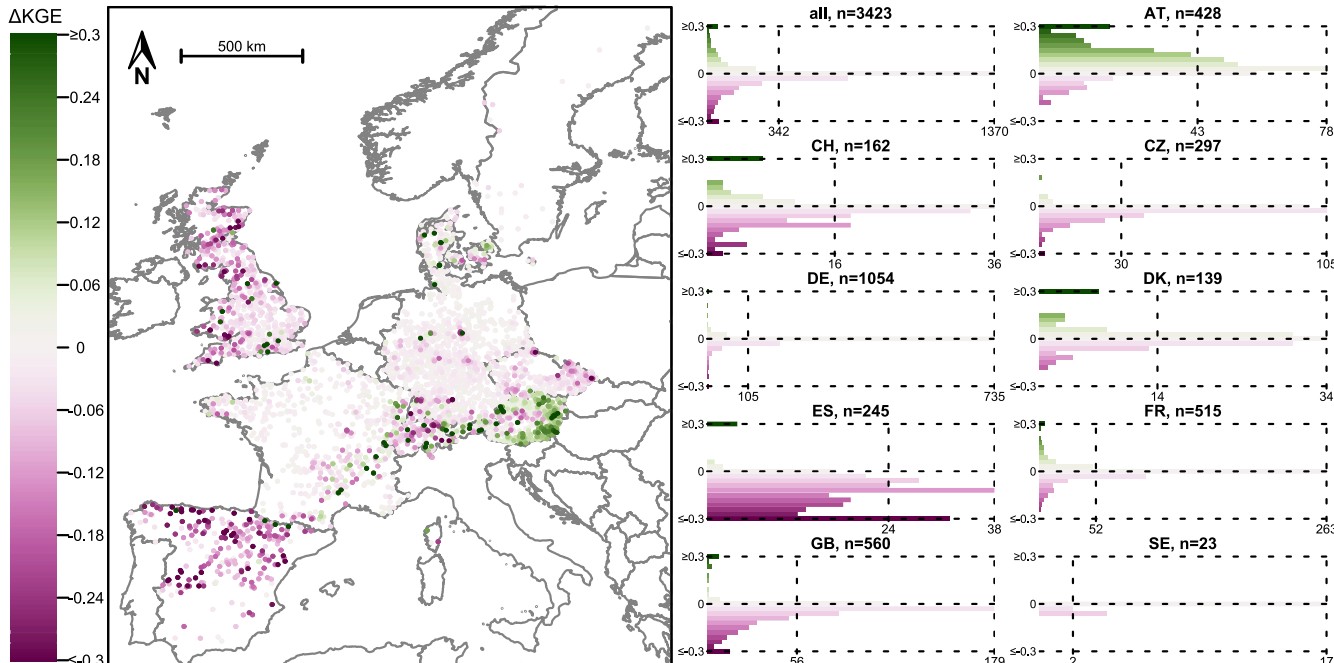

**Figure A4: Difference in model performance when all meteorological input data were obtained from E-OBS (i.e., EStreams, scenario II) and when the precipitation data from E-OBS were replaced with those from CAMELS (scenario III). Positive values and green colours indicate higher model performances with the precipitation data from E-OBS, negative values and pink colours indicate higher model performances with the precipitation data from CAMELS. Note that the colour scale was cut at a difference in KGE of ±0.3. The catchments with the largest differences in model performance were plotted last to increase their visibility.**






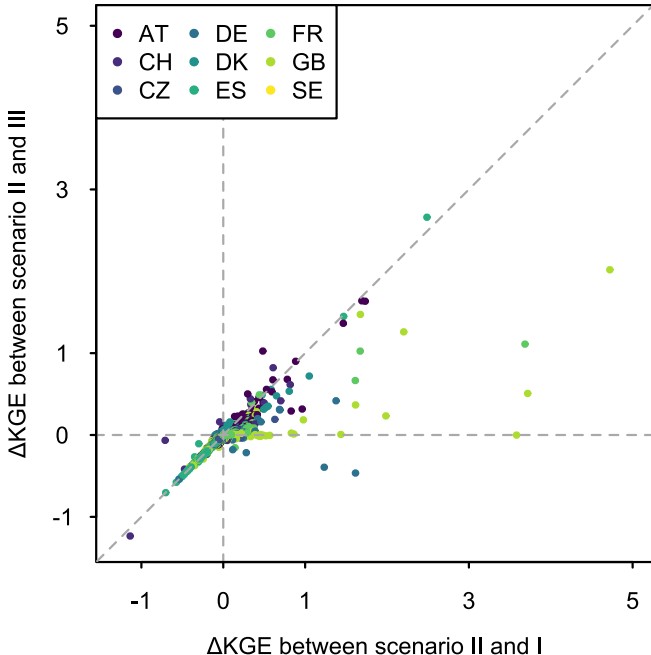

**Figure A5: Differences in model performance between scenario II and I (positive values indicate higher performances with the E-OBS data obtained from EStreams, negative values indicate higher performances with the CAMELS data, see Fig. 6) compared to differences in model performance between scenarios II and III (positive values indicate higher performances with the E-OBS data, negative values indicate higher performances when the precipitation data were replaced with those from CAMELS, see Fig. A4). One catchment (in Great Britain) plotted outside the axis limits (11.1 / 5.5). Pearson's correlation coefficient was 0.86.**





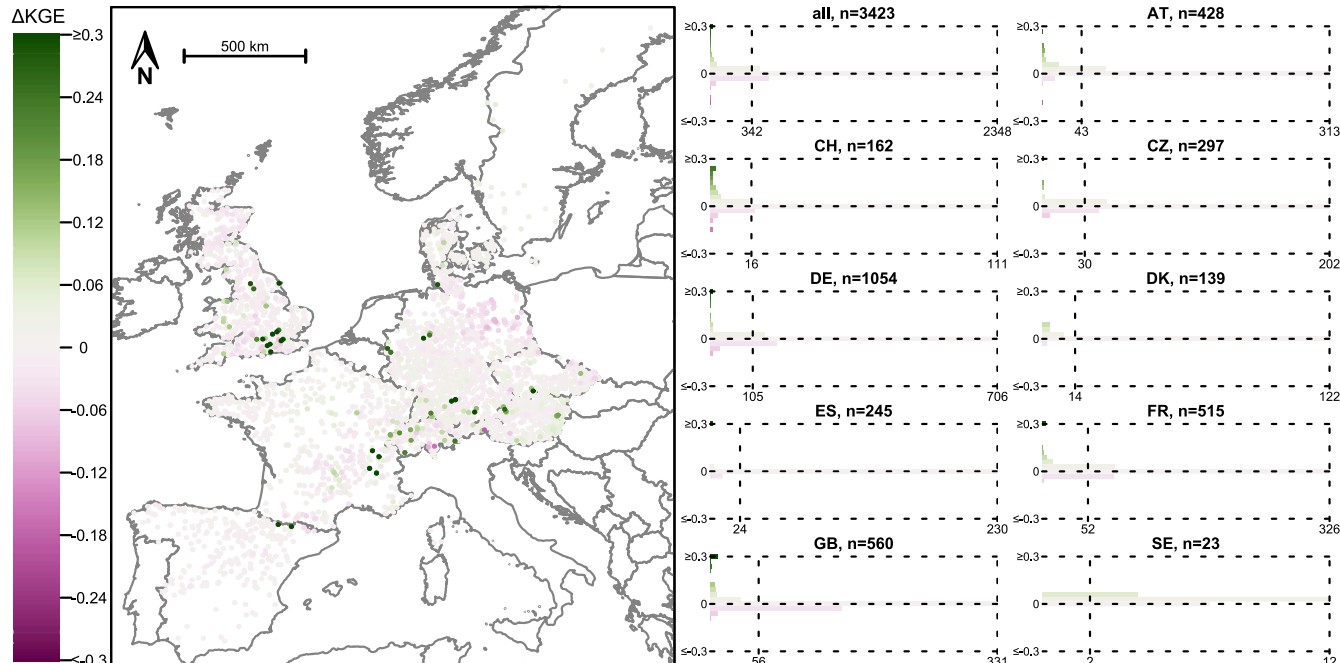

**Figure A6: Difference in model performance when all meteorological input data were obtained from E-OBS (i.e., EStreams, scenario II) and when the $E_{pot}$ data from E-OBS were replaced with those from CAMELS (scenario IV). Positive values and green colours indicate higher model performances with the $E_{pot}$ data from E-OBS, negative values and pink colours indicate higher model performances with the $E_{pot}$ data from CAMELS. Note that the colour scale was cut at a difference in KGE of ±0.3. The catchments with**
**the largest differences in model performance were plotted last to increase their visibility.**



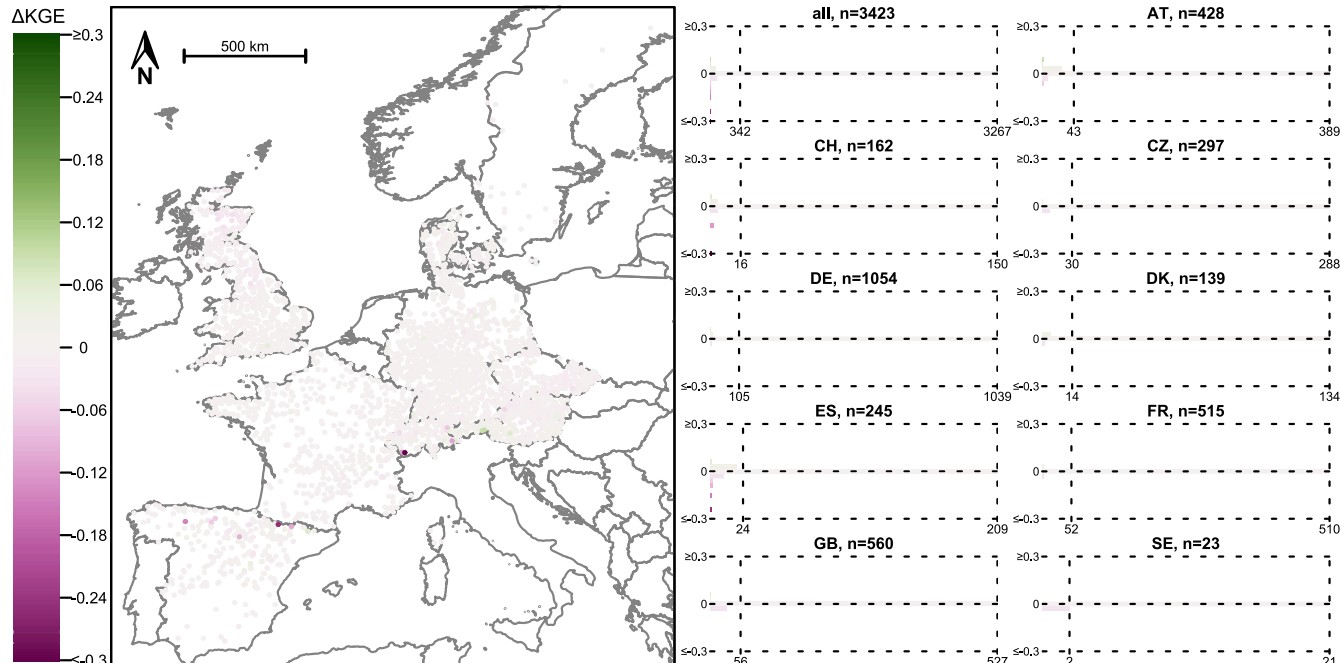

**Figure A7: Difference in model performance when all meteorological input data were obtained from E-OBS (i.e., EStreams, scenario II) and when the temperature data from E-OBS were replaced with those from CAMELS (scenario V). Positive values and green colours indicate higher model performances with the *T* data from E-OBS, negative values and pink colours indicate higher model performances with the *T* data from CAMELS. Note that the colour scale was cut at a difference in KGE of ±0.3. The catchments with the largest differences in model performance were plotted last to increase their visibility.**






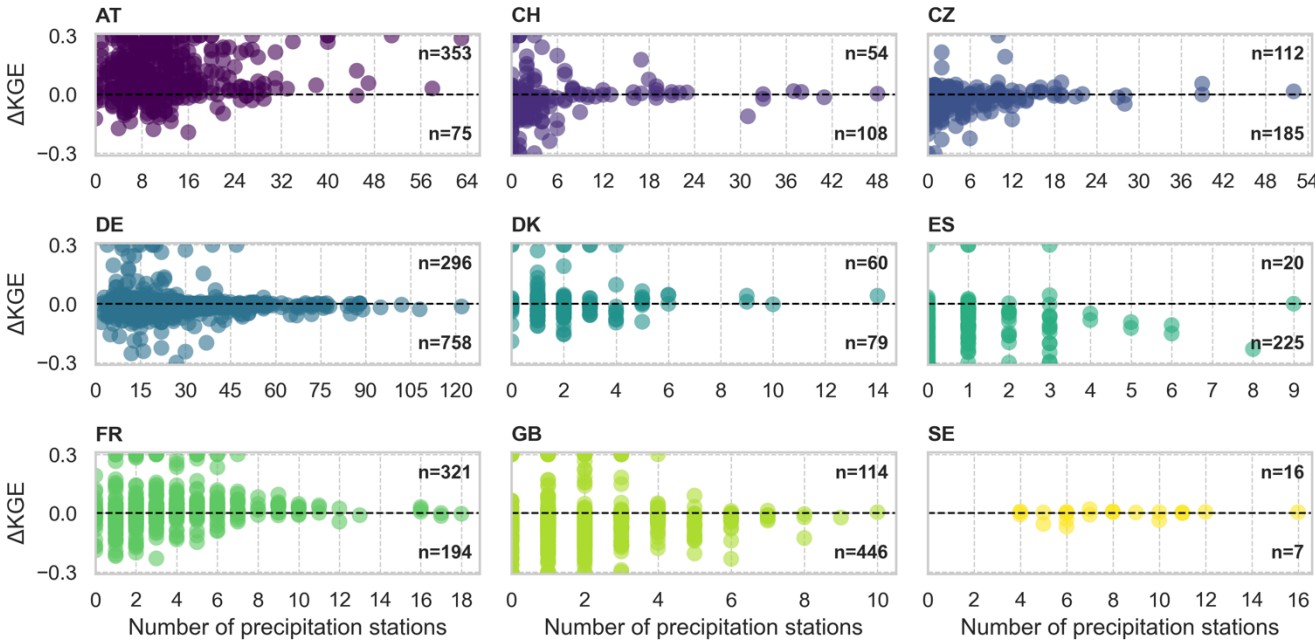

**Figure A8: Scatterplots showing the difference in model performance (Kling-Gupta efficiency, KGE) between scenario II (E-OBS data obtained from EStreams) and scenario I (CAMELS data) (y-axis) versus the number of precipitation stations used to derive the E-OBS precipitation data per country. Each circle represents one catchment. Positive values indicate higher performances when the E-OBS data were used, negative values indicate higher performances when the CAMELS data were used. Note that the y-axes were cut at ±0.3, in accordance to Fig. 6. Note that the x-axes differ for the different subplots.**