# Peer review of "Evaluating E-OBS forcing data for large-sample hydrology using model performance diagnostics"

_EGUsphere, 2025_

## Referee Comment (RC2)

**Manuscript review: "Evaluating the quality of the E-OBS meteorological forcing data in EStreams for large-sample hydrology studies in Europe"**

This paper compares the E-OBS meteorological forcing data used in the pan-European EStreams dataset to the meteorological forcing data of nine regional datasets. As such, it provides useful insights into the suitability of EStreams for large-sample hydrology applications. The results show that precipitation in EStreams is generally lower than in the regional datasets, while temperature and potential evapotranspiration are higher. Hydrological model performance is typically slightly lower with EStreams than with the regional forcing datasets. The paper is clearly written, and the dataset comparison is valuable to the LSH community. However, some aspects related to the methodology, interpretation and presentation can be improved (see comments below).

**Major comments**

**1. Framing of "quality assessment"**

The title of the paper indicates that a direct evaluation of the quality of the E-OBS meteorological forcing data is performed, e.g. through comparison to meteorological station observations. However, the aim of the paper is to compare E-OBS forcing data in EStreams to meteorological forcings from regional datasets, which themselves may still contain errors and biases. As stated in the introduction, rather than a quality assessment, the study evaluates the "overall hydrological efficacy of the meteorological forcing data". Clarifying this distinction in the title would better align expectations with the actual scope of the work.

**2. Use of KGE as the sole performance metric**

The performance of the hydrological model is evaluated with the Kling–Gupta Efficiency (KGE). While KGE is widely used, it depends strongly on flow variability and can therefore mislead when used as the only performance indicator in a large-sample study such as this one. Because important conclusions are drawn from spatial differences in KGE (e.g. higher model performance in wetter catchments), I recommend including at least one complementary error-based metric (e.g. RMSE, NRMSE, or percent bias) to better distinguish between variability effects and true model accuracy. See e.g. Williams, 2025.

**3. Influence of basin regulation and anthropogenic impacts**

The paper briefly acknowledges human influence in some catchments (e.g. dams, diversions), but this is not reflected in the data selection criteria. The EStreams dataset includes information about dams and total upstream reservoir volume. Additionally, for some of the regional datasets used (e.g. BULL in Spain and LamaH-CE in Austria), further

information on the degree of human impact is also available. I strongly recommend using this metadata to exclude regulated or heavily influenced basins in Section 2.1 ("Subset of catchments") wherever possible. Even if human influence affects hydrological model simulations across all forcing products (scenarios) similarly, removing impacted basins would increase the robustness and interpretability of the comparison.

For example, in the Discussion (lines 333–336), the manuscript notes that the low model performance in Spain *may* be linked to human influence. Excluding impacted basins where metadata is available (such as in the BULL dataset) would help clarify whether regulation is indeed a substantial contributor to the lower model performances in those regions.

**4. Differences between forcing products**

I miss a discussion on the different types of the local datasets (e.g. observation-based, reanalysis-based, ..). These types have different strengths and limitations depending on factors such as terrain complexity and station density, which may contribute to regional performance differences. Also, it would be interesting to know how much overlap there is in the source data between the EStreams and CAMELS forcings.

Further, some regional forcing datasets have considerably higher spatial resolution (e.g. 1–5 km) than E-OBS (0.25°), yet the implications of these differences are not discussed. A short discussion of whether resolution differences contribute to the observed spatial patterns would strengthen the interpretation.

**Minor comments:**

A major finding is that the mean annual precipitation sums in the E-OBS data are lower than in the regional datasets. The potential reasons for this are not discussed anywhere. Possible reasons for this could be discussed in the Discussion section (4.1).

Line 91-93: Please explain why gauges with average streamflow above 10 mm/d were omitted, as well as gauges with runoff ratio above 1.1.

Also, a map that shows which of the candidate basins were eliminated in which filter step would be helpful (visualize section 2.1). It would show why a certain EStreams basin is not included in the final analysis. This should be easy to make, but as-is, the description of the selection filter is not all that informative. The map could go in the appendix or supplemental material.

Line 108: Please remove the text in line 108 beginning with "EStreams is a ready-to-use product..." through to "...for the evaluation of the E-OBS meteorological data.", as it does not add information beyond what has already been stated.

The following sentence "Note that there is also a version of E-OBS at a resolution of 0.1° available, but not represented in EStreams." can instead be moved to directly follow the sentence that starts with "In EStreams..." in line 103. In addition, it would be helpful to briefly explain why the 0.1° E-OBS version was not used in EStreams, as this is not addressed in the original EStreams paper.

Line 120: I suggest adding a column to Table 2 to specify what type of dataset in each case (e.g. observation-based, reanalysis-based, ..)

Line 142: Specify the spatial resolution of the DEM

Line 262: The sentence "For the catchments in the center of Austria, the CAMELS data sometimes led to better model performances than the E-OBS data, while the opposite was the case in most other catchments (see above)." Can be removed.

Line 294: Add (Fig. 7) to the end of the sentence.

Line 298: Figure A8 is not discussed anywhere in the manuscript. Consider removing it or adding a brief interpretation of its relevance.

Line 361: consider replacing the word "striking" (e.g. with "considerable")

Line 364: This text is confusing: "The Epot calculations for each catchment in EStreams with the Hargreaves equation (do Nascimento et al., 2024) thus also affected the resulting Epot data that we used to represent the E-OBS Epot. However, the Hargreaves equation was found to be reliable, among other regions especially in Central Europe (Pimentel et al., 2023) and this choice can therefore be supported"

I suggest replacing it with something like: "Epot calculated with the Hargreaves equation, as in Estreams, has been found to be reliable e.g. in Central Europe (Pimental et al., 2023)."

Additionally, since you state "among other regions", consider citing other references that support the use of the Hargreaves equation in other regions.

Line 383: Consider stating earlier in the manuscript (e.g. in Sect. 2: Data and Methods) that the methodology of this study is based on the study by Clerc-Schwarzenbach et al. (2024).

**Technical corrections**

Line 47: Change "arised" to "arose"

Line 348: If using "previous studies," cite more than one source or change to singular.

**References**

Williams, G. P.: Friends don't let friends use Nash-Sutcliffe Efficiency (NSE) or KGE for hydrologic model accuracy evaluation: A rant with data and suggestions for better practice, Environmental Modelling & Software, 194, 106665, https://doi.org/10.1016/J.ENVSOFT.2025.106665, 2025.

---

## Author Comment (AC1)

Dear Reviewer 1, dear Alex

Thank you very much for your encouraging and detailed review of our manuscript. Your comments will be very helpful for further improvements. Please find below our replies to the review comments and how we will implement them in the revised version of the paper. We used *blue italic font* to distinguish the comments from our replies. Of course, we will also implement the technical corrections. Thank you for spotting them.

Best wishes,

Franziska Clerc-Schwarzenbach & Thiago do Nascimento

**L11:**

"limitations of data quality" -> maybe indicate that data quality is expected to vary in space? (e.g. "limitations and regional variations of data quality")

We thank the reviewer for the feedback. We will modify the text to: "limitations and regional variations of data quality."

**L36-L39:**

The number of catchments is not directly the problem, the mixture of different regions / countries is the challenge, as meteorological data is often available on a national level (e.g. provided by national meteorological organizations)

We thank the reviewer for this input. We will adjust the introduction to make sure that this is explicitly stated and to avoid that the number of catchments is stated to be the problem.

**L40, L45, L55, L72, L417...:**

I know what you want to say here, but I don't like the word "standardization" in this context, as it usually refers to something else when it comes to data processing, and e.g. ERA5 or E-OBS are just datasets on a larger scale with different sources and processing methods, they do not "standardize" smaller datasets.

We thank the reviewer for raising this point. After careful consideration, we will change the wording at all instances, to make sure that we point to the consistency of the data over large spatial extents without using the potentially misleading word "standardization".

**L91-L93:**

**Why did you apply these criteria?**

Catchments with an average streamflow above 10 mm/day were excluded since such values exceed typical ranges reported in LSH datasets (normally < 5 mm/day) by far, and may indicate data inconsistencies (e.g., overestimated streamflow or underestimated area), or glacier-dominated hydrology.

Catchments with runoff ratios above 1.1 were removed because natural runoff rarely exceeds precipitation by large margins, and such instances could indicate data errors or strong human influence.

We will add our reasonings for these constraints in section 2.1.

**L111-L112:**

The resolution of 0.25° of E-OBS is very coarse, I know that e.g. the precipitation data for CAMELS-DE has a resolution of 1x1 km, this could be an additional source for limitations of Estreams data, also for comparisons in this study. Maybe you could think about an update of Estreams in the future? I think this could be worth it (not part of this study).

We thank the reviewer for the feedback and input. In the meantime, EStreams is being updated with the E-OBS data with a 0.1° resolution. We will thus rerun all the model runs and redo the analysis with the data with a 0.1° resolution. Preliminary results show that this will not change our results strongly, still we think it is fair to include the data with the highest resolution available. However, we will include in section 2.2 that the different spatial resolutions of the CAMELS and the E-OBS datasets are expected to lead to different performances.

**L116-L117:**

I think the main thing here is that the quality and uncertainty of E-OBS data have a larger (regional) spread, some regions will have very good quality data (where station measurements are available), other regions with less station measurements will have worse data quality. Even if the data comes from the same source (E-OBS), quality and uncertainty varies regionally. I think this is a major challenge in LSH and people need to be aware of this.

We thank the reviewer for these thoughts and the valuable discussion. We will include these differences already here and will stress that using the same dataset in two different regions does not necessarily imply the same data quality for both regions. We would like also to point out that this is further discussed in section 4.2.

**L149:**

Maybe add a small explanation on why you designed the scenarios this way, and which questions you aim to answer with the different scenarios (I and II are quite clear, but why did you do III-V?)

We thank the reviewer for pointing out that this is unclear. We also believe that this point touches some of the discussion in the comment from the EGU peer review training (CC1).

Scenarios III-V were chosen to disentangle the impact of each forcing time series from E-OBS in the results driven by scenario II. In other words:

- Scenario III was chosen to evaluate the impact of precipitation.
- Scenario IV was chosen to evaluate the impact of  $E_{pot}$ , and consequently to evaluate whether using a different  $E_{pot}$  formulation would change our main results. (See also comment about the possibility of using different  $E_{pot}$  formulations by the reviewer from the peer review training, i.e., CC1.)
- Scenario V was chosen to evaluate the impact of temperature.

We will update section 2.4 with explanations on the motivation behind each of the scenarios, by extending the statement in the very beginning of the section.

**L152:**

This could also go into limitations, but the catchment shapes are also not identical between EStreams and the CAMELS datasets, which results in different areas for which the meteorological data was "cut out" and aggregated, which can also lead to differences.

We thank the reviewer for raising this point. We will include a remark on this issue in section 2.2 to make sure that readers (and, more importantly, users of the datasets) are aware of this.

**L312:**

I think it is hard to see any patterns in this figure with the mixture of scenario I and II with circles and triangles. I am not sure on how to improve the figure, but you could calculate a regression line and also report the p-values? This could also be used to back up your statement in L310-311

We thank the reviewer for suggesting these helpful improvements to the figure. We will modify the figure by including the correlation between the two variables (Spearman ranking coefficient and p-value) and additionally plot the lowess (locally weighted) smooth line for the trend assessment for each subplot.

**L324-L325:**

results in Austria are bad as ERA5 data is used, not a "local" dataset, maybe add this here?

We thank the reviewer for calling our attention to the missing remark on the special issue of Austria at this instance. As suggested by Reviewer 3, we will exclude the results for Austria from the study to avoid including a dataset that fundamentally differs from the others (i.e., that is not a national dataset). We will note however that the LamaH-CE dataset is different to make sure that this point still comes across and users are aware of it.

**L344-L353:**

Here you have the paragraph about limitations of ERA5 data in Austria, but I think it does not really fit in the paragraph ("Evaluation of the E-OBS data in comparison to the E-OBS station density"). Maybe it could fit better in Section 4.1? I think the point about ERA5 data used in Austria is very important in this study, as this is a fundamental difference to the other CAMELS datasets, where local, highest-quality data is used, in Austria it is quite the opposite. You should make this point very clear, also in the beginning, as you do not test whether "local" CAMELS data is better than E-OBS data in the case of Austria.

We thank the reviewer for this comment. As in the new version, we will only add a statement about Austria and why we did not use it in the study, we think that this problem is solved (while still making sure that the message comes across).

It would also be interesting to see how the different CAMELS precipitation data was collected / processed (maybe not so easy to find out). I only know about CAMELS-DE, but

HYRAS is also based on interpolated station data (I guess mostly the same stations as used for E-OBS), which would explain the relative similarities, but it is still interesting to see that there are differences (maybe due to different interpolation / processing methods or the coarser resolution of E-OBS)

We believe that this is a valuable remark, and thank the reviewer for it. Following the suggestion of Reviewer 2, we will include information on the origins of the data in Table 2. In addition, we will add a section in the discussion in order to discuss the potential implications of the different data origins.

**L398-L401:**

For smaller catchments, having E-OBS data from the 0.1° version could also help (again, maybe this is worth an update for EStreams, which of course is not part of this study, just a general suggestion)

We thank the reviewer for sharing this thought. In the meantime, EStreams is being made available with forcing data from E-OBS at a 0.1° spatial resolution. Thus, we will rerun all the model runs and redo the analysis with the highest resolution data.

**L402...:**

You could add to the conclusion that local datasets are usually the best, but using E-OBS data and EStreams offer a great harmonized data source for LSH studies covering all of Europe, especially as an alternative to ERA5 which has shown limitations in Austria. Maybe extend a little bit on this and how E-OBS could be an alternative to ERA5 which was mostly the standard before.

We thank the reviewer for the suggestion to enrich the conclusions. We will modify the last paragraph of the conclusions to something in those lines:

"Overall, while local or nationally curated datasets often yield the best model performances due to their finer spatial resolution and denser station networks, our results suggest that the meteorological forcing E-OBS data in EStreams represents a valuable and harmonized alternative for pan-European studies. The advantage of E-OBS lies in its observational basis, consistent methodology, and coverage across all of Europe, making it especially useful when national datasets are unavailable or inconsistent. As such, E-OBS and EStreams provide a practical foundation for expanding large-sample hydrology beyond

national boundaries while maintaining sufficient data quality for robust model applications."

---

## Author Comment (AC2)

**Dear Reviewer 2**

Thank you very much for your encouraging and detailed review of our manuscript. Your comments will be very helpful in further improving the study. Please find below our replies to the review comments and how we will implement them in the revised version of the paper. We used *blue italic font* to distinguish the comments from our replies. Of course, we will also implement the technical corrections. Thank you for making us aware of them.

Best regards,

Franziska Clerc-Schwarzenbach & Thiago do Nascimento

**Major comments**

**1. Framing of "quality assessment"**

The title of the paper indicates that a direct evaluation of the quality of the E-OBS meteorological forcing data is performed, e.g. through comparison to meteorological station observations. However, the aim of the paper is to compare E-OBS forcing data in EStreams to meteorological forcings from regional datasets, which themselves may still contain errors and biases. As stated in the introduction, rather than a quality assessment, the study evaluates the "overall hydrological efficacy of the meteorological forcing data". Clarifying this distinction in the title would better align expectations with the actual scope of the work.

We thank the reviewer for pointing out this weakness about our manuscript title and for the suggestions for improvement. We agree that the current use of the term "quality" in the title might be misleading, and following insights also brought by Reviewer 3, we will change the title accordingly.

**2. Use of KGE as the sole performance metric**

The performance of the hydrological model is evaluated with the Kling–Gupta Efficiency (KGE). While KGE is widely used, it depends strongly on flow variability and can therefore mislead when used as the only performance indicator in a large-sample study such as this one. Because important conclusions are drawn from spatial differences in KGE (e.g. higher model performance in wetter catchments), I recommend including at least one complementary error-based metric (e.g. RMSE, NRMSE, or percent bias) to better distinguish between variability effects and true model accuracy. See e.g. Williams, 2025.

We thank the reviewer for their insights and the valuable hint to current literature. We will keep the KGE as a performance measure to be able to compare the different scenarios (and because people are used to interpreting it). We will though take care not to draw conclusions based on solely the comparison of the KGE *between* catchments, to avoid the dependency on flow variability and catchment area, for example. We will additionally use the percent bias (PBIAS) as a complementary error-based metric to improve the discussion of variability effects.

**3. Influence of basin regulation and anthropogenic impacts**

The paper briefly acknowledges human influence in some catchments (e.g. dams, diversions), but this is not reflected in the data selection criteria. The EStreams dataset includes information about dams and total upstream reservoir volume. Additionally, for some of the regional datasets used (e.g. BULL in Spain and LamaH-CE in Austria), further information on the degree of human impact is also available. I strongly recommend using this metadata to exclude regulated or heavily influenced basins in Section 2.1 ("Subset of catchments") wherever possible. Even if human influence affects hydrological model simulations across all forcing products (scenarios) similarly, removing impacted basins would increase the robustness and interpretability of the comparison. For example, in the Discussion (lines 333–336), the manuscript notes that the low model performance in Spain may be linked to human influence. Excluding impacted basins where metadata is available (such as in the BULL dataset) would help clarify whether regulation is indeed a substantial contributor to the lower model performances in those regions.

We thank the reviewer for raising this important point. After discussion, we decided to incorporate some information on lakes and reservoirs available in EStreams into our filtering procedure. Specifically, we now will retain only catchments that meet the following criteria:

- Number of lakes upstream < 5</li>
- Normalized upstream capacity < 0.2</li>

The normalized upstream capacity was computed following Salwey et al. (2023), and the threshold value was defined based on their findings. We acknowledge that this filter may exclude some catchments that are not substantially affected in their water balance by regulation; however, we chose to adopt a more conservative (stricter) filtering approach, aiming to exclude all heavily regulated catchments.

Furthermore, we opted to consider only the number of lakes and the normalized upstream capacity because this information is readily available in EStreams, ensuring a consistent and fair filtering process across countries, even where anthropogenic impacts are not explicitly indicated in their CAMELS-like datasets.

We will include this information in the cascade of exclusion criteria in section 2.1.

Salwey, S., Coxon, G., Pianosi, F., Singer, M. B., & Hutton, C. (2023). National-scale detection of reservoir impacts through hydrological signatures. Water Resources Research, 59, e2022WR033893. https://doi.org/10.1029/2022WR033893

**4. Differences between forcing products**

I miss a discussion on the different types of the local datasets (e.g. observation-based, reanalysis-based, ..). These types have different strengths and limitations depending on factors such as terrain complexity and station density, which may contribute to regional performance differences. Also, it would be interesting to know how much overlap there is in the source data between the EStreams and CAMELS forcings. Further, some regional forcing datasets have considerably higher spatial resolution (e.g. 15 km) than E-OBS (0.25°), yet the implications of these differences are not discussed. A short discussion of whether resolution differences contribute to the observed spatial patterns would strengthen the interpretation.

We thank the reviewer for bringing up this very important point. Following this recommendation and also the suggestion of Reviewer 1, we will add information on the different origins of forcing data in Table 2 and include a section where we discuss the potential implications of these differences in the discussion.

**Minor comments**

A major finding is that the mean annual precipitation sums in the E-OBS data are lower than in the regional datasets. The potential reasons for this are not discussed anywhere. Possible reasons for this could be discussed in the Discussion section (4.1).

We will add potential reasons for the clearly lower precipitation data in E-OBS to section 4.1.

Line 91-93: Please explain why gauges with average streamflow above 10 mm/d were omitted, as well as gauges with runoff ratio above 1.1.

Catchments with an average streamflow above 10 mm/day were excluded, as such values significantly exceed typical ranges reported in LSH datasets, and may indicate data inconsistencies (e.g., overestimated streamflow or underestimated area) or glacier-dominated hydrology.

Catchments with runoff ratios above 1.1 were removed because natural runoff rarely exceeds precipitation by large margins, and such instances could indicate data errors or strong human influence.

Also, in accordance with Reviewer 1, we will add the motivations behind these exclusion criteria in the cascade given in section 2.1.

Also, a map that shows which of the candidate basins were eliminated in which filter step would be helpful (visualize section 2.1). It would show why a certain EStreams basin is not included in the final analysis. This should be easy to make, but as-is, the description of the selection filter is not all that informative. The map could go in the appendix or supplemental material.

We thank the reviewer for this suggestion. We will add a figure with these maps in the appendix.

Line 108: Please remove the text in line 108 beginning with "EStreams is a ready-to-use product..." through to "...for the evaluation of the E-OBS meteorological data.", as it does not add information beyond what has already been stated.

We will remove the sentence as suggested.

The following sentence "Note that there is also a version of E-OBS at a resolution of 0.1° available, but not represented in EStreams." can instead be moved to directly follow the sentence that starts with "In EStreams..." in line 103. In addition, it would be helpful to briefly explain why the 0.1° E-OBS version was not used in EStreams, as this is not addressed in the original EStreams paper.

We thank the reviewer for raising this issue. In the meantime, EStreams is being updated with the E-OBS data at a 0.1° resolution. We will thus rerun the model runs and redo the analyses for the new data obtained from EStreams. Therefore, we believe that this problem is solved. Preliminary results indicate that there will only be slight differences to the current results.

Line 120: I suggest adding a column to Table 2 to specify what type of dataset in each case (e.g. observation-based, reanalysis-based, ..)

We thank the reviewer for the suggestion and will modify Table 2 accordingly.

Line 142: Specify the spatial resolution of the DEM

The resolution used was 30 m. We will add this information to L142.

Line 262: The sentence "For the catchments in the center of Austria, the CAMELS data sometimes led to better model performances than the E-OBS data, while the opposite was the case in most other catchments (see above)." Can be removed.

We will remove the sentence as suggested by the reviewer, as the Austrian catchments will no longer be part of the study: Based on a suggestion by Reviewer 3, we decided to exclude the LamaH-CE dataset because it is substantially different from the other CAMELS datasets. We will state the reason for this exclusion in section 2.1.

Line 294: Add (Fig. 7) to the end of the sentence.

We will do this.

Line 298: Figure A8 is not discussed anywhere in the manuscript. Consider removing it or adding a brief interpretation of its relevance.

We will remove it.

**Line 361: consider replacing the word "striking" (e.g. with "considerable")**

We will replace the word accordingly.

Line 364: This text is confusing: "The Epot calculations for each catchment in EStreams with the Hargreaves equation (do Nascimento et al., 2024) thus also affected the resulting Epot data that we used to represent the E-OBS Epot. However, the Hargreaves equation was found to be reliable, among other regions especially in Central Europe (Pimentel et al., 2023) and this choice can therefore be supported" I suggest replacing it with something like: "Epot calculated with the Hargreaves equation, as in Estreams, has been found to be reliable e.g. in Central Europe (Pimental et al., 2023)."

We thank the reviewer for the suggestion. We will modify the sentence to something along the lines:

" $E_{\rm pot}$  calculated with the Hargreaves equation, as in EStreams, has been found to be a reliable method in various hydrological modelling applications, including in Central Europe (Pimental et al., 2023), and other regions (Weiland et al., 2012; Bangi and Soraganvi, 2023)."

Weiland, F., Tisseuil, C., Dürr, H., Vrac, M., & van Beek, L. (2012). Selecting the optimal method to calculate daily global reference potential evaporation from CFSR reanalysis data for application in a hydrological model study. Hydrology and Earth System Sciences, 16, 983–1000.

Bangi, S. C., & Soraganvi, V. S. (2023). *A modified temperature-based model for estimation of potential evapotranspiration over Ghataprabha river basin, South India*. Spatial Information Research, 31, 583–595.

Additionally, since you state "among other regions", consider citing other references that support the use of the Hargreaves equation in other regions.

Please see the reply to the comment above.

Line 383: Consider stating earlier in the manuscript (e.g. in Sect. 2: Data and Methods) that the methodology of this study is based on the study by Clerc-Schwarzenbach et al. (2024).

We will make this sentence into Section 2.

---

## Author Comment (AC3)

**Dear Reviewer 3**

Many thanks for reviewing our manuscript. Your remarks will be very helpful in further improving the study. Please find below our replies to the review comments and how we will implement them in the revised version of the paper. We used *blue italic font* to distinguish the comments from our replies.

Best regards,

Franziska Clerc-Schwarzenbach & Thiago do Nascimento

**Major remarks**

First, your paper would deserve a better title. As I understood, you ask a much more general and (from my point of view) interesting question: how can we use a classical precipitation-runoff model such as HBV in order to compare the quality of precipitation data. I believe you should put this point at the forefront of your paper. You should discuss the "good sense" (almost philosophical) hypothesis of your approach: even if your hydrological model is imperfect, the difference of efficiency when calibrating the model with different forcings cannot be due to some random factor. Better performances cannot be due to chance. You could perhaps look at this chapter of the famous Ray Linsley (1982) who discussed the topic, I only remember this short citation "if the data are too poor for the use of a good simulation model they are also inadequate for any other model", but there must be some other interesting citations there.

We thank the reviewer for bringing this up and for the valuable suggestion for literature. We think that the paper will benefit from putting more emphasis on how we use the hydrological model and why we think that it is a meaningful way to do so. We will thus include this in the introduction. In addition, we will change the title to put a spotlight on the methodology already there (see also comment by Reviewer 2).

Second, I believe that it is worth comparing the CAMELS outputs with the E-obs outputs, introducing a further class of inputs (ERA-5) makes things more complex. I would simply have discarded the LAMAH dataset, stating that you aim at comparing the "best ground-based estimate" of the CAMELS datasets with the E-obs... it is definitely not a big surprise that ERA-5 estimates are not good... and it makes your paper unnecessarily more complex. You do not have to show us everything you have done, if you have pushed open at a few open doors in the course of your research (what we all do...) you do not need to tell us about it.

We thank the reviewer for encouraging us to exclude the Austrian catchments from the analysis. In fact, much of the presentation of the results and the discussion would be simpler if the special case of LamaH-CE was not included in the study. Thus, we decided to include a short note on what makes the LamaH-CE dataset different from the others and why we thus did not use it for this study in section 2.1 and exclude these catchments from further analyses.

Third, I was wondering whether it would have been interesting to restrict the dataset to the less-regulated (reservoir-impacted) catchments. I know for example that there are quite a few regulated catchments in the Swiss CAMELS dataset. It will not change the results, but a focus on the less regulated catchments could perhaps show even clearer differences.

We thank the reviewer for raising this issue. Based on this recommendation as well as the recommendation by Reviewer 2, we decided to exclude catchments with a normalized upstream capacity larger than 0.2 (Salwey et al., 2023) as well as catchments with 5 or more lakes upstreams. We use these criteria as they are available in the EStreams dataset and can thus be used for all catchments of the study consistently.

We will include this information in the cascade of exclusion criteria in section 2.1.

Salwey, S., Coxon, G., Pianosi, F., Singer, M. B., & Hutton, C. (2023). National-scale detection of reservoir impacts through hydrological signatures. Water Resources Research, 59, e2022WR033893. <a href="https://doi.org/10.1029/2022WR033893">https://doi.org/10.1029/2022WR033893</a>

**Minor comments**

I believe that before mentioning (I.36) that "the inclusion of an increasing number of catchments in one dataset almost always goes hand in hand with difficulties in providing high-quality forcing data" you should underline that large samples also come with their load of problematic discharge stations. In my experience of building a CAMELS dataset, a large part of the effort was absorbed by scrutinizing collectively the time series, the locations, etc. And because E-streams did not make any sorting, there must be along with the hydrometric stations a few (or more) non sense stations (probably a few buoys in France...) or at least stations which measure a level that cannot be related to any significant hydrological flux.

We thank the reviewer for addressing this potential issue. We will underline along line 36 that larger large-sample datasets are expected to be more prone to wrong streamflow data than smaller large-sample datasets that were sorted and filtered by hand. However, since we only

use catchments that are included in one of the CAMELS datasets (where we assume that filtering took place in all cases), we believe that only meaningful stations were included in our study. Furthermore, since we used the streamflow data from the CAMELS datasets in all scenarios (see next comment), we do not expect any issues regarding wrong or inconsistent streamflow stations.

**Did you check that the discharge data were exactly the same in E-stream and CAMELS?**

For streamflow, we used the data provided in the different CAMELS datasets for all scenarios, for two reasons: a) to make sure that differences in model performances were due to the meteorological input data and not affected by potential differences in the streamflow data, and b) since EStreams only provides information on how to get to the streamflow data of the different stations, but not streamflow data directly. We will add this information in section 2.4 and are thankful for the remark on this that made us aware that this information is currently missing.

In 3.3.1 (Number of E-OBS precipitation stations): I believe you should mention that the number of E-Obs stations is correlated with the size of the catchments... and as you (and all the conceptual modelers) know, the largest catchments get the best KGE criteria.

We thank the reviewer for raising this point. We will include a remark on this dependency, as well as further information regarding the relationship between catchment areas and numbers of E-OBS stations at this place in the manuscript.

---

## Author Comment (AC4)

Dear Ryan Teuling

Thank you for providing us with the review from the reviewer training.

Dear Reviewer from the peer review training

Thank you for choosing our manuscript and for your comments on it. Please find our replies to your comments below. We used *blue italic font* to distinguish the comments from our replies.

Best regards,

Franziska Clerc-Schwarzenbach & Thiago do Nascimento

**Main comments**

Regarding the methods, the authors have included all catchments in their studies, regardless of being impacted by human activities or not (L94-97). This can be questionable, as only the climate forcings are used as the hydrological model inputs. This modeling approach may only be applicable to the natural sites without human intervention. Among the 3423 catchments, these include much noise in the modeling results. Importantly, this approach makes it hard to differentiate if one type of meteorological data is better than the other one because of natural condition or human intervention or the quality (or station density) of the meteorological data itself.

We thank the reviewer for raising this important point. After discussion, we decided to incorporate some information on lakes and reservoirs available in EStreams into our filtering procedure. Specifically, we now will retain only catchments that meet the following criteria:

- Number of lakes upstream < 5</li>
- Normalized upstream capacity < 0.2</li>

The normalized upstream capacity was computed following Salwey et al. (2023), and the threshold value was defined based on their findings. We acknowledge that this filter may exclude some catchments that are not substantially affected in their water balance by regulation; however, we chose to adopt a more conservative (stricter) filtering approach, aiming to exclude all heavily regulated catchments.

Furthermore, we opted to consider only the number of lakes and the normalized upstream capacity because this information is readily available in EStreams, ensuring a consistent and fair filtering process across countries, even where anthropogenic impacts are not explicitly indicated in their CAMELS-like datasets.

We will include this information in the cascade of exclusion criteria in section 2.1.

Salwey, S., Coxon, G., Pianosi, F., Singer, M. B., & Hutton, C. (2023). National-scale detection of reservoir impacts through hydrological signatures. Water Resources Research, 59, e2022WR033893. https://doi.org/10.1029/2022WR033893

What about other possible governing factors, such as climate types, topography, land use and land cover, and geology? These are all not addressed or analyzed by coming to the conclusion due to spatial resolution and station density. Simply saying one is better than the other without analyzing the possible governing factors could limit the applicability and generalization of the research outputs. Therefore, more analysis on process-based understanding and transferable knowledge is needed to make robust conclusions supported by the evidence.

Climate, topography, land use, land cover, and geology all remain constant, independent of the meteorological forcings used to calibrate a model (i.e., in the different scenarios, the only factor that we changed are the meteorological forcings). Thus, we would argue that the catchment characteristics are not relevant for the evaluation of the hydrological efficacy of the meteorological forcings when used for a hydrological model calibration.

The authors adopted the potential evapotranspiration data derived from different approaches: it is calculated with the simplest approach (only temperature based) in E-OBS, but with different varieties of methods in the CAMELS. If the authors want to do a comparison, it should be "apple" to "apple". It is recommended that the potential evapotranspiration should be calculated with the same methods for both types of datasets.

For this study, we used different existing datasets. These datasets are openly available and contain data that are ready to be used. Having said that, it is most likely that a user of the dataset will make use of those potential evapotranspiration data that are provided in the dataset of interest. As the different datasets (i.e., the different CAMELS and CAMELS-like datasets as well as the EStreams dataset) were created by different teams and for different

regions, it is in the nature of the subject that different approaches to calculate potential evapotranspiration were used.

We argue that a comparison of the results (in this case: the potential evapotranspiration data) is especially relevant when different approaches (in this case: different equations to calculate potential evapotranspiration) were used with the same goal. After all, it is in the interest of the modeler to know how different the input data is depending on which dataset they choose to gather data for a certain catchment. Thus, we consider it valuable to not change the data that were made available in the different datasets, but to work with what is provided (and is thus used by the community).

Regarding the results, it would be more useful to state the governing factors (climatology, topography, land use, etc.) why E-OBS has over- or underestimations compared with CAMELS, besides simply stating which countries or regions have higher or lower meteorological values. More exactly, why one dataset is better than the other one in some countries yes while some countries not?

We thank the reviewer for making us aware that this occurs to be incomplete. We checked for correlations of our results with other catchment characteristics and did not find anything besides what we stated. We will include a sentence in the results section making this clear. Furthermore, as suggested by Reviewer 2, we will include a discussion on potential reasons for the differences in the precipitation data.

Another key aspect is that the authors calibrate the models individually with different climate datasets. Therefore, not only the climate data are different, but the model parameters are different. Therefore, the model performance lower or higher is not only due to climate data quality but also the model parameters.

The 'optimal' parametrization of a bucket-type hydrological model may differ depending on the meteorological input data (for example, if the model tries to compensate for a bias in the data). We do not see any possibility of making a fair comparison of the model performances without informing the model with the different meteorological input data that it has to deal with then. Furthermore, note that for each model performance value, we used ten independently optimized parametrizations to avoid a strong dependence from one parameterization.

**Specific comments**

L10: Maybe mentioned the spatial resolution of the meteorological data from the E-OBS?

While this is surely important information, we do not think that it should be part of the abstract for which the length is limited. All information on the spatial resolution of the E-OBS data is given in section 2.2. Note that in the next version, we will use the E-OBS data at a resolution of 0.1° as these is being made available in EStreams in the meantime.

L16: Model performance is SLIGHTLY lower when E-OBS data are used compared with CAMELS data: is this difference statistically significant?

We thank the reviewer for this question. To evaluate whether the difference in model performance between E-OBS and CAMELS forcing data is statistically significant, we conducted a Wilcoxon signed-rank test (Wilcoxon, 1945) on the paired KGE values. The test indicates that the median KGE for CAMELS (0.883) is slightly higher than for E-OBS (0.867), and that this difference is statistically significant (Wilcoxon signed-rank test: W =  $3.59 \times 10^6$ , p <  $10^{-29}$ ). Although the difference is statistically significant due to the large sample size, the effect size is small, suggesting that the practical difference in model performance is minor. We will add this information to the manuscript. (Note however that the numbers may change due to the change in spatial resolution of the E-OBS forcing data).

Wilcoxon, F.: Individual comparisons by ranking methods, Biometrics Bull., 1, 80–83, 1945.

L48-53: the authors actually come to the same conclusion as the referred literature, and mentioned the same thing in the abstract. So what is the added value of evaluating E-OBS vs. CAMELS? Just because of a larger scale of detailed dataset?

As stated in L53-54: "Yet, evaluations of the E-OBS data for a larger extent, and specifically for hydrological modelling, remain unexplored." – So far, there have been no tests of the E-OBS data in a hydrological model, and especially not in a comparison to alternative data. For a hydrological modeler working on large-sample hydrology in Europe, this study will support an informed decision for (or against) a certain dataset.

L84: Why exclude the catchments with area more than 2000 km2? What is the impact or relation between the catchment area and the meteorological data?

We thank the reviewer for this question, pointing out that a statement on the motivation for this decision is missing. For a large catchment system (arbitrary threshold of 2000 km²), a bucket-type hydrological model may not be the most suitable choice. Therefore, we excluded catchments larger than that from this study. We will add a sentence clarifying this in the cascade of criteria in section 2.1.

L123-132: Why are the annual differences of precipitation and evapotranspiration between the datasets compared but not the seasonal differences? While for temperature, you compared the daily differences?

We are aware that the comparisons we made only provide a limited picture of the differences in the meteorological input data. However, as this is not a study purely on data comparison, we decided to include one measure per variable. For temperature, it is not possible to calculate an annual sum that can be compared. Therefore, the mean daily difference (which is the same as when the annual mean temperature is calculated first, and the mean difference is calculated then) is given in that case.

Figure 4: What are the reasons for the different model performance among the countries? What are the governing factors? Simply stating the KGE is higher here or lower there without providing further reasons sounds not helpful.

We thank the reviewer for their comments. However, note that in the section "Results" we only describe the results of the study, without interpretation. The reasons for the higher or lower model performances – as far as they could be identified – are given in the section "Discussion". Motivated by a comment of Reviewer 2, we will avoid comparing the model performances (i.e., the KGE values) for different regions between each other.

Figure 6: The important thing is not the exact number of catchments in a country where E-OBS dataset is better or worse than the CAMELS datasets, but why E-OBS is better/worse than the CAMELS in these catchments?

We thank the reviewer for pointing this out. We think that it can be helpful to know which meteorological input data lead to a more successful streamflow simulation, since the model performance can be interpreted as an aggregated measure for hydrological efficacy (and thus gives an indication of which data may be of a higher quality). Regarding the reasons for the lower or higher model performances, these are discussed in the section "Discussion", where the modelling results are interpreted.

L275-278: Why is the model performance lower in Great Britain which shows opposite behavior? Please explain.

In scenario III, we use the (higher) potential evapotranspiration data from EStreams, while in scenario II, we use the (lower) potential evapotranspiration data from CAMELS (in this case, CAMELS-GB). In section 4.3, we explain why for the karstic catchments in Great Britain, the higher potential evapotranspiration data were beneficial. We will add a sentence at lines 275-278 indicating that for these catchments, potential evapotranspiration was more important than elsewhere to make the link to the explanation provided later.

Figure 7: Simply stating the station density plays the key role seems not convincing, as the author stated that other factors may also play a role. It would be more interesting to analyze other factors as well? Are the relationships between the station numbers and the KGE statistically significant?

We have computed the correlations between KGE and catchment descriptors, and this statement is based on that. Specifically, only the correlation with the number of precipitation (and temperature) stations achieved a statistically significant coefficient.

We agree that there is room for improvement in the text and results, and we will add the table with the correlations in the appendix. We will also improve the discussion and also make some improvements in the figure, such as including the Spearman ranking coefficient (relationships) and p-values (significance) for each subplot (country).

Figure 8: What about a trend assessment on the data? Is there a significant relationship between model performance and aridity index?

We believe that this is a fair assessment, and agree that it is currently lacking. We will compute the correlation between the two variables (Spearman ranking coefficient and p-value) and consequently plot the LOWESS (locally weighted) smooth line for the trend assessment for each subplot in Figure 8.

L366-369: it is too assertive and not supported by evidence. It is a very simple method to calculate the potential evapotranspiration which does not consider solar radiation impact. It is also too assertive to say different calculation approaches of potential evapotranspiration will not change the results.

Note that the evidence that the Hargreaves equation is reliable (e.g., in Central Europe) does not origin from the current study, but from the study by Pimentel et al. (2023), cited in this sentence. Furthermore, the choice for the Hargreaves equation was not made for this study, but when the EStreams dataset was published. The statement that the different potential evapotranspiration data did not affect the model performance results strongly is supported by Figure A6.

To address this comment, we will change L366-369 to:

"Furthermore, the differences in  $E_{pot}$  data did not affect model performance results strongly (as can be seen in Figure A6), so it can be expected that the use of a different equation would not change the findings of this study."

Additionally, we will add two other references supporting the reliability of the Hargreaves, as suggested by Reviewer 2.

Weiland, F., Tisseuil, C., Dürr, H., Vrac, M., & van Beek, L. (2012). Selecting the optimal method to calculate daily global reference potential evaporation from CFSR reanalysis data for application in a hydrological model study. Hydrology and Earth System Sciences, 16, 983–1000.

Bangi, S. C., & Soraganvi, V. S. (2023). *A modified temperature-based model for estimation of potential evapotranspiration over Ghataprabha river basin, South India*. Spatial Information Research, 31, 583–595.

---

## Author Response (AR1)

Dear Editor, dear Albrecht Weerts

Hereby, we are submitting the revised version of our manuscript. We implemented all the changes that we promised to the reviewers during the discussion. In the replies to the referees, we list how and where the changes were made.

We also changed the conclusions in the way you asked us to, thank you for this remark.

Please note that we changed the title of our manuscript and that we decided to have supporting information to not overload the appendix.

We are looking forward to your feedback and thank you for your time and support.

Best wishes,

Franziska Clerc-Schwarzenbach & Thiago do Nascimento

Dear Reviewer 1, dear Alex

Thank you very much for your encouraging and detailed review of our manuscript. Your comments were very helpful for further improvements. Please find below our replies to the review comments and how we implemented them in the revised version of the paper. We used *blue italic font* to distinguish the comments from our replies. Of course, we also implemented the technical corrections. Thank you for spotting them.

Best wishes,

Franziska Clerc-Schwarzenbach & Thiago do Nascimento

*L11:*
*"limitations of data quality" -> maybe indicate that data quality is expected to vary in space? (e.g. "limitations and regional variations of data quality")*

We thank the reviewer for the feedback. We have modified the text to:

L11"limitations and regional variations of data quality."

*L36-L39:*
*The number of catchments is not directly the problem, the mixture of different regions / countries is the challenge, as meteorological data is often available on a national level (e.g. provided by national meteorological organizations)*

We thank the reviewer for this input. We have adjusted the introduction (L34-38) to make sure that this is explicitly stated and to avoid that the number of catchments is stated to be the challenge.

*L40, L45, L55, L72, L417...:*
*I know what you want to say here, but I don't like the word "standardization" in this context, as it usually refers to something else when it comes to data processing, and e.g. ERA5 or E-OBS are just datasets on a larger scale with different sources and processing methods, they do not "standardize" smaller datasets.*

We thank the reviewer for raising this point. We have changed the wording at all instances to words as "large-scale", "harmonized", etc., to make sure that we point to the

consistency of the data over large spatial extents without using the potentially misleading word "standardization".

*Why did you apply these criteria?*

Catchments with an average streamflow above 10 mm/day were excluded since such values exceed typical ranges reported in LSH datasets (normally < 5 mm/day) by far, and may indicate data inconsistencies (e.g., overestimated streamflow or underestimated area), or glacier-dominated hydrology. Note that with the new catchment selection, the average streamflow filter did not lead to an exclusion of any catchments anymore.

Catchments with runoff ratios above 1.1 were removed because natural runoff rarely exceeds precipitation by large margins, and such instances could indicate data errors or strong human influence.

We have added such reasonings along L107-122 in section 2.1.

*The resolution of 0.25° of E-OBS is very coarse, I know that e.g. the precipitation data for CAMELS-DE has a resolution of 1x1 km, this could be an additional source for limitations of Estreams data, also for comparisons in this study. Maybe you could think about an update of Estreams in the future? I think this could be worth it (not part of this study).*

In the meantime, EStreams has being updated with the E-OBS data with a 0.1° resolution, and we have rerun all the model runs and redid the analysis with the data with a 0.1° resolution.

However, we have also included in section 2.2 that the different spatial resolutions of a forcing data are expected to lead to different performances (L138).

*I think the main thing here is that the quality and uncertainty of E-OBS data have a larger (regional) spread, some regions will have very good quality data (where station measurements are available), other regions with less station measurements will have worse data quality. Even if the data comes from the same source (E-OBS), quality and*

*uncertainty varies regionally. I think this is a major challenge in LSH and people need to be aware of this.*

We thank the reviewer for these thoughts and the valuable discussion. We have included these differences already here and have stressed that using the same dataset in two different regions does not necessarily imply the same data quality for both regions in discussion 4.3.

*L149:*
*Maybe add a small explanation on why you designed the scenarios this way, and which questions you aim to answer with the different scenarios (I and II are quite clear, but why did you do III-V?)*

We have updated section 2.4 with explanations on the motivation behind each of the scenarios, and by extending the statement in the very beginning of the section.

*L152:*
*This could also go into limitations, but the catchment shapes are also not identical between EStreams and the CAMELS datasets, which results in different areas for which the meteorological data was "cut out" and aggregated, which can also lead to differences.*

We thank the reviewer for raising this point. We have included a remark on this issue in section 2.2 (L145-146) to make sure that readers (and, more importantly, users of the datasets) are aware of this.

*L312:*
*I think it is hard to see any patterns in this figure with the mixture of scenario I and II with circles and triangles. I am not sure on how to improve the figure, but you could calculate a regression line and also report the p-values? This could also be used to back up your statement in L310-311*

We have modified the figure by including the correlation between the two variables (Spearman rank coefficient and *p*-value) and additionally plot the lowess (locally weighted) smooth line for the trend assessment for each subplot. The figure description and discussions have also been updated accordingly.

*L324-L325:*
*results in Austria are bad as ERA5 data is used, not a "local" dataset, maybe add this here?*

As suggested by Reviewer 3, we have excluded the results for Austria from the study to avoid including a dataset that fundamentally differs from the others (i.e., that is not a national dataset).

*L344-L353:*
*Here you have the paragraph about limitations of ERA5 data in Austria, but I think it does not really fit in the paragraph ("Evaluation of the E-OBS data in comparison to the E-OBS station density"). Maybe it could fit better in Section 4.1? I think the point about ERA5 data used in Austria is very important in this study, as this is a fundamental difference to the other CAMELS datasets, where local, highest-quality data is used, in Austria it is quite the opposite. You should make this point very clear, also in the beginning, as you do not test whether "local" CAMELS data is better than E-OBS data in the case of Austria.*

We thank the reviewer for this comment. As in the new version, we have only added a statement in section 2.1 (L109-112) about Austria and why we did not use it in the study.

*It would also be interesting to see how the different CAMELS precipitation data was collected / processed (maybe not so easy to find out). I only know about CAMELS-DE, but HYRAS is also based on interpolated station data (I guess mostly the same stations as used for E-OBS), which would explain the relative similarities, but it is still interesting to see that there are differences (maybe due to different interpolation / processing methods or the coarser resolution of E-OBS)*

We believe that this is a valuable remark and thank the reviewer for it. Following the suggestion of Reviewer 2, we have included information on the origins of the data in Table 2. In addition, we have added a section in the discussion to discuss the potential implications of the different data origins, characteristics and underlying station density (Section 4.3).

*L398-L401:*
*For smaller catchments, having E-OBS data from the 0.1° version could also help (again, maybe this is worth an update for EStreams, which of course is not part of this study, just a general suggestion)*

We thank the reviewer for sharing this thought. In the meantime, EStreams has being made available with forcing data from E-OBS at a 0.1° spatial resolution. Thus, have rerun all the model runs and redone the analysis with the highest resolution data.

*L402...:*
*You could add to the conclusion that local datasets are usually the best, but using E-OBS data and EStreams offer a great harmonized data source for LSH studies covering all of Europe, especially as an alternative to ERA5 which has shown limitations in Austria. Maybe extend a little bit on this and how E-OBS could be an alternative to ERA5 which was mostly the standard before.*

We have modified the last paragraph of the conclusions (L501-506):

"Overall, while local or national datasets often yield the best model performances, our results suggest that the meteorological forcing data from E-OBS that is included in EStreams represents a valuable and harmonized alternative for pan-European studies. The advantage of E-OBS lies in its observational basis, consistent methodology, and coverage across all of Europe, making it especially useful when national datasets are unavailable or inconsistent. As such, E-OBS and EStreams provide a practical foundation for expanding large-sample hydrology beyond national boundaries while maintaining sufficient data quality for robust model applications."

Dear Reviewer 2

Thank you very much for your encouraging and detailed review of our manuscript. Your comments were very helpful in improving the study. Please find below our replies to the review comments and how we implemented them in the revised version of the paper. We used *blue italic font* to distinguish the comments from our replies. Of course, we also implemented the technical corrections. Thank you for making us aware of them.

Best regards,

Franziska Clerc-Schwarzenbach & Thiago do Nascimento

**Major comments**

*1. Framing of "quality assessment"*
*The title of the paper indicates that a direct evaluation of the quality of the E-OBS meteorological forcing data is performed, e.g. through comparison to meteorological station observations. However, the aim of the paper is to compare E-OBS forcing data in EStreams to meteorological forcings from regional datasets, which themselves may still contain errors and biases. As stated in the introduction, rather than a quality assessment, the study evaluates the "overall hydrological efficacy of the meteorological forcing data". Clarifying this distinction in the title would better align expectations with the actual scope of the work.*

We thank the reviewer for pointing out this weakness about our manuscript title and for the suggestions for improvement. We have changed the tile to: "Evaluating E-OBS forcing data for large-sample hydrology using model performance diagnostics"

*2. Use of KGE as the sole performance metric*
*The performance of the hydrological model is evaluated with the Kling–Gupta Efficiency (KGE). While KGE is widely used, it depends strongly on flow variability and can therefore mislead when used as the only performance indicator in a large-sample study such as this one. Because important conclusions are drawn from spatial differences in KGE (e.g. higher model performance in wetter catchments), I recommend including at least one complementary error-based metric (e.g. RMSE, NRMSE, or percent bias) to better distinguish between variability effects and true model accuracy. See e.g. Williams, 2025.*

We thank the reviewer for their insights and the valuable hint to current literature. We have kept the KGE as a performance measure to be able to compare the different scenarios (and

because people are used to interpreting it). We have though taken care not to draw conclusions based on solely the comparison of the KGE *between* catchments, to avoid the dependency on flow variability and catchment area, for example. We have additionally used the percent bias (PBIAS) as a complementary error-based metric to improve the discussion of variability effects, as can be seen along L265-271 and in the newly added Fig. 5 and Fig. A4.

*3. Influence of basin regulation and anthropogenic impacts*
*The paper briefly acknowledges human influence in some catchments (e.g. dams, diversions), but this is not reflected in the data selection criteria. The EStreams dataset includes information about dams and total upstream reservoir volume. Additionally, for some of the regional datasets used (e.g. BULL in Spain and LamaH-CE in Austria), further information on the degree of human impact is also available. I strongly recommend using this metadata to exclude regulated or heavily influenced basins in Section 2.1 ("Subset of catchments") wherever possible. Even if human influence affects hydrological model simulations across all forcing products (scenarios) similarly, removing impacted basins would increase the robustness and interpretability of the comparison. For example, in the Discussion (lines 333–336), the manuscript notes that the low model performance in Spain may be linked to human influence. Excluding impacted basins where metadata is available (such as in the BULL dataset) would help clarify whether regulation is indeed a substantial contributor to the lower model performances in those regions.*

We thank the reviewer for raising this important point. We have incorporated some information on lakes and reservoirs available in EStreams into our filtering procedure (Section 2.1), specifically the number of lakes upstream and the normalized upstream capacity of reservoirs and have added the needed explanation along L113-119.

*4. Differences between forcing products*
*I miss a discussion on the different types of the local datasets (e.g. observation-based, reanalysis-based, ..). These types have different strengths and limitations depending on factors such as terrain complexity and station density, which may contribute to regional performance differences. Also, it would be interesting to know how much overlap there is in the source data between the EStreams and CAMELS forcings. Further, some regional forcing datasets have considerably higher spatial resolution (e.g. 1 5 km) than E-OBS (0.25°), yet the implications of these differences are not discussed. A short discussion of*

*whether resolution differences contribute to the observed spatial patterns would strengthen the interpretation.*

We thank the reviewer for bringing up this very important point. Following this recommendation and also the suggestion of Reviewer 1, we have added information on the different origins of forcing data in Table 2 and also included Section 4.3, where we discussed the potential implications of these differences among the CAMELS and E-OBS datasets.

**Minor comments**

*A major finding is that the mean annual precipitation sums in the E-OBS data are lower than in the regional datasets. The potential reasons for this are not discussed anywhere. Possible reasons for this could be discussed in the Discussion section (4.1).*

We have added a discussion (also previous literature) about potential reasonings in the newly added section 4.1.

*Line 91-93: Please explain why gauges with average streamflow above 10 mm/d were omitted, as well as gauges with runoff ratio above 1.1.*

We have added the motivations behind these exclusion criteria in the cascade given in section 2.1, specifically, L120-122.

*Also, a map that shows which of the candidate basins were eliminated in which filter step would be helpful (visualize section 2.1). It would show why a certain EStreams basin is not included in the final analysis. This should be easy to make, but as-is, the description of the selection filter is not all that informative. The map could go in the appendix or supplemental material.*

We have added Figure S1 with these maps in the Supporting Information.

*Line 108: Please remove the text in line 108 beginning with "EStreams is a ready-to-use product..." through to "...for the evaluation of the E-OBS meteorological data.", as it does not add information beyond what has already been stated.*

We have removed the sentence as suggested.

*The following sentence "Note that there is also a version of E-OBS at a resolution of 0.1° available, but not represented in EStreams." can instead be moved to directly follow the sentence that starts with "In EStreams..." in line 103. In addition, it would be helpful to briefly explain why the 0.1° E-OBS version was not used in EStreams, as this is not addressed in the original EStreams paper.*

We thank the reviewer for raising this issue. In the meantime, EStreams has being updated with the E-OBS data at a 0.1° resolution, and the manuscript has updated the analysis with the new resolution.

*Line 120: I suggest adding a column to Table 2 to specify what type of dataset in each case (e.g. observation-based, reanalysis-based, ..)*

We thank the reviewer for the suggestion, and we have modified Table 2 accordingly.

*Line 142: Specify the spatial resolution of the DEM*

We have added this information to L179 (30 m).

*Line 262: The sentence "For the catchments in the center of Austria, the CAMELS data sometimes led to better model performances than the E-OBS data, while the opposite was the case in most other catchments (see above)." Can be removed.*

We have removed the sentence as suggested by the reviewer, as the Austrian catchments are no long part of the study.

*Line 294: Add (Fig. 7) to the end of the sentence.*

We have done this.

*Line 298: Figure A8 is not discussed anywhere in the manuscript. Consider removing it or adding a brief interpretation of its relevance.*

We have removed it.

*Line 361: consider replacing the word "striking" (e.g. with "considerable")*

We have replaced the word accordingly.

*Line 364: This text is confusing: "The Epot calculations for each catchment in EStreams with the Hargreaves equation (do Nascimento et al., 2024) thus also affected the resulting Epot data that we used to represent the E-OBS Epot. However, the Hargreaves equation was found to be reliable, among other regions especially in Central Europe (Pimentel et al., 2023) and this choice can therefore be supported" I suggest replacing it with something like: "Epot calculated with the Hargreaves equation, as in Estreams, has been found to be reliable e.g. in Central Europe (Pimental et al., 2023)."*

We thank the reviewer for the suggestion. We have modified the sentence accordingly at L447-450.

*Additionally, since you state "among other regions", consider citing other references that support the use of the Hargreaves equation in other regions.*

Please see the reply to the comment above.

*Line 383: Consider stating earlier in the manuscript (e.g. in Sect. 2: Data and Methods) that the methodology of this study is based on the study by Clerc-Schwarzenbach et al. (2024).*

We have added such statement in both, Introduction (L67) and in Section 2.4 (L163).

Dear Reviewer 3

Many thanks for reviewing our manuscript. Your remarks were very helpful for improving the study. Please find below our replies to the review comments and how we implemented them in the revised version of the paper. We used *blue italic font* to distinguish the comments from our replies.

Best regards,

Franziska Clerc-Schwarzenbach & Thiago do Nascimento

**Major remarks**

*First, your paper would deserve a better title. As I understood, you ask a much more general and (from my point of view) interesting question: how can we use a classical precipitation-runoff model such as HBV in order to compare the quality of precipitation data. I believe you should put this point at the forefront of your paper. You should discuss the "good sense" (almost philosophical) hypothesis of your approach: even if your hydrological model is imperfect, the difference of efficiency when calibrating the model with different forcings cannot be due to some random factor. Better performances cannot be due to chance. You could perhaps look at this chapter of the famous Ray Linsley (1982) who discussed the topic, I only remember this short citation "if the data are too poor for the use of a good simulation model they are also inadequate for any other model", but there must be some other interesting citations there.*

We thank the reviewer for bringing this up and for the valuable suggestion for literature.

We have included the reference (which fits perfectly with our hypothesis) in the introduction, which was improved (L72-79). In addition, we have changed the title to put a spotlight on the methodology already there (see also comment by Reviewer 2) to:

"Evaluating E-OBS forcing data for large-sample hydrology using model performance diagnostics"

*Second, I believe that it is worth comparing the CAMELS outputs with the E-obs outputs, introducing a further class of inputs (ERA-5) makes things more complex. I would simply have discarded the LAMAH dataset, stating that you aim at comparing the "best ground-*

*based estimate" of the CAMELS datasets with the E-obs... it is definitely not a big surprise that ERA-5 estimates are not good... and it makes your paper unnecessarily more complex. You do not have to show us everything you have done, if you have pushed open at a few open doors in the course of your research (what we all do...) you do not need to tell us about it.*

We thank the reviewer for encouraging us to exclude the Austrian catchments from the analysis. In fact, much of the presentation of the results and the discussion would be simpler if the special case of LamaH-CE was not included in the study. Thus, have included a short note on what makes the LamaH-CE dataset different from the others and why we thus did not use it for this study in section 2.1 and excluded these catchments from further analyses.

*Third, I was wondering whether it would have been interesting to restrict the dataset to the less-regulated (reservoir-impacted) catchments. I know for example that there are quite a few regulated catchments in the Swiss CAMELS dataset. It will not change the results, but a focus on the less regulated catchments could perhaps show even clearer differences.*

We thank the reviewer for raising this issue. Based on this recommendation as well as the recommendation by Reviewer 2, we have added two new filters: catchments with a normalized upstream capacity larger than 0.2 (Salwey et al., 2023) as well as catchments with 5 or more lakes upstream. These changes are described in Section 2.1 and L99-101 and L113-119.

**Minor comments**

*I believe that before mentioning (l.36) that "the inclusion of an increasing number of catchments in one dataset almost always goes hand in hand with difficulties in providing high-quality forcing data" you should underline that large samples also come with their load of problematic discharge stations. In my experience of building a CAMELS dataset, a large part of the effort was absorbed by scrutinizing collectively the time series, the locations, etc. And because E-streams did not make any sorting, there must be along with the hydrometric stations a few (or more) non sense stations (probably a few buoys in France...) or at least stations which measure a level that cannot be related to any significant hydrological flux.*

We have underlined along L34-39 that larger large-sample datasets are expected to be more prone to wrong streamflow data than smaller large-sample datasets that were sorted and filtered by hand.

However, since we only use catchments that are included in one of the CAMELS datasets (where we assume that filtering took place in all cases), we believe that only meaningful stations were included in our study. Furthermore, since we used the streamflow data from the CAMELS datasets in all scenarios (see next comment), we do not expect any issues regarding wrong or inconsistent streamflow stations.

*Did you check that the discharge data were exactly the same in E-stream and CAMELS?*

For streamflow, we used the data provided in the different CAMELS datasets for all scenarios, for two reasons: a) to make sure that differences in model performances were due to the meteorological input data and not affected by potential differences in the streamflow data, and b) since EStreams only provides information on how to get to the streamflow data of the different stations, but not streamflow data directly. We have added this information in section 2.4 (L165-167) and are thankful for the remark on this that made us aware that this information was missing.

*In 3.3.1 (Number of E-OBS precipitation stations): I believe you should mention that the number of E-Obs stations is correlated with the size of the catchments... and as you (and all the conceptual modelers) know, the largest catchments get the best KGE criteria.*

We thank the reviewer for raising this point. We have both included a remark on this dependency, as well as further information regarding the relationship between catchment areas and numbers of E-OBS stations in section 3.3.1.

Dear Ryan Teuling

Thank you for providing us with the review from the reviewer training.

Dear Reviewer from the peer review training

Thank you for choosing our manuscript and for your comments on it. Please find our replies to your comments below. We used *blue italic font* to distinguish the comments from our replies.

Best regards,

Franziska Clerc-Schwarzenbach & Thiago do Nascimento

**Main comments**

*Regarding the methods, the authors have included all catchments in their studies, regardless of being impacted by human activities or not (L94-97). This can be questionable, as only the climate forcings are used as the hydrological model inputs. This modeling approach may only be applicable to the natural sites without human intervention. Among the 3423 catchments, these include much noise in the modeling results. Importantly, this approach makes it hard to differentiate if one type of meteorological data is better than the other one because of natural condition or human intervention or the quality (or station density) of the meteorological data itself.*

We thank the reviewer for raising this important point. We have incorporated some information on lakes and reservoirs available in EStreams into our filtering procedure (Section 2.1), specifically the number of lakes upstream and the normalized upstream capacity of reservoirs and have added the needed explanation along L113-119.

*What about other possible governing factors, such as climate types, topography, land use and land cover, and geology? These are all not addressed or analyzed by coming to the conclusion due to spatial resolution and station density. Simply saying one is better than the other without analyzing the possible governing factors could limit the applicability and generalization of the research outputs. Therefore, more analysis on process-based*

*understanding and transferable knowledge is needed to make robust conclusions supported by the evidence.*

Climate, topography, land use, land cover, and geology all remain constant, independent of the meteorological forcings used to calibrate a model (i.e., in the different scenarios, the only factor that we changed are the meteorological forcings). Thus, we would argue that the catchment characteristics are not relevant for the evaluation of the hydrological efficacy of the meteorological forcings when used for a hydrological model calibration.

*The authors adopted the potential evapotranspiration data derived from different approaches: it is calculated with the simplest approach (only temperature based) in E-OBS, but with different varieties of methods in the CAMELS. If the authors want to do a comparison, it should be "apple" to "apple". It is recommended that the potential evapotranspiration should be calculated with the same methods for both types of datasets.*

For this study, we used different existing datasets. These datasets are openly available and contain data that are ready to be used. Having said that, it is most likely that a user of the dataset will make use of those potential evapotranspiration data that are provided in the dataset of interest. As the different datasets (i.e., the different CAMELS and CAMELS-like datasets as well as the EStreams dataset) were created by different teams and for different regions, it is in the nature of the subject that different approaches to calculate potential evapotranspiration were used.

We argue that a comparison of the results (in this case: the potential evapotranspiration data) is especially relevant when different approaches (in this case: different equations to calculate potential evapotranspiration) were used with the same goal. After all, it is in the interest of the modeler to know how different the input data is depending on which dataset they choose to gather data for a certain catchment. Thus, we consider it valuable to not change the data that were made available in the different datasets, but to work with what is provided (and is thus used by the community).

In order to bring awareness of the underlying choices, we have added and improved the discussion, specifically Section 4.1, 4.2 and 4.3, which touch the concerns raised.

*Regarding the results, it would be more useful to state the governing factors (climatology, topography, land use, etc.) why E-OBS has over- or underestimations compared with*

*CAMELS, besides simply stating which countries or regions have higher or lower meteorological values. More exactly, why one dataset is better than the other one in some countries yes while some countries not?*

We thank the reviewer for making us aware that this occurs to be incomplete. We checked for correlations of our results with other catchment characteristics and did not find anything noteworthy besides what we stated. We have included a sentence in the results section making this clear (3.3), alongside the found correlations (Table S1). Furthermore, as suggested by Reviewer 2, we have included section 4.1 listing potential reasons for the differences in the precipitation data.

*Another key aspect is that the authors calibrate the models individually with different climate datasets. Therefore, not only the climate data are different, but the model parameters are different. Therefore, the model performance lower or higher is not only due to climate data quality but also the model parameters.*

The 'optimal' parametrization of a bucket-type hydrological model may differ depending on the meteorological input data (for example, if the model tries to compensate for a bias in the data). We do not see any possibility of making a fair comparison of the model performances without informing the model with the different meteorological input data that it has to deal with then. Furthermore, note that for each model performance value, we used ten independently optimized parametrizations to avoid a strong dependence from one parameterization.

**Specific comments**

*L10: Maybe mentioned the spatial resolution of the meteorological data from the E-OBS?*

While this is surely important information, we do not think that it should be part of the abstract for which the length is limited. All information on the spatial resolution of the E-OBS data is given in section 2.2. Note that in the current version, we have used the E-OBS data at a resolution of 0.1° as these has been made available in EStreams in the meantime.

*L16: Model performance is SLIGHTLY lower when E-OBS data are used compared with CAMELS data: is this difference statistically significant?*

We thank the reviewer for this question. To evaluate whether the difference in model performance between E-OBS and CAMELS forcing data is statistically significant, we conducted a Wilcoxon signed-rank test (Wilcoxon, 1945) on the paired KGE values. The test indicates that the median KGE for CAMELS is slightly higher than for E-OBS, and that this difference is statistically significant ($p$-value < 0.001). Although the difference is statistically significant due to the large sample size, the effect size is small, suggesting that the practical difference in model performance is minor. We have added this information to the manuscript (L258-259).

*L48-53: the authors actually come to the same conclusion as the referred literature, and mentioned the same thing in the abstract. So what is the added value of evaluating E-OBS vs. CAMELS? Just because of a larger scale of detailed dataset?*

As stated in the introduction: "Yet, evaluations of the E-OBS data for a larger extent, and specifically for hydrological modelling, remain unexplored." – So far, there have been no tests of the E-OBS data in a hydrological model, and especially not in a comparison to alternative data. For a hydrological modeler working on large-sample hydrology in Europe, this study will support an informed decision for (or against) a certain dataset.

*L84: Why exclude the catchments with area more than 2000 km2? What is the impact or relation between the catchment area and the meteorological data?*

For a large catchment system (arbitrary threshold of 2000 km$^2$), a bucket-type hydrological model may not be the most suitable choice. Therefore, we excluded catchments larger than that from this study. We have added a sentence clarifying this right below the cascade of criteria in section 2.1 (L107-108).

*L123-132: Why are the annual differences of precipitation and evapotranspiration between the datasets compared but not the seasonal differences? While for temperature, you compared the daily differences?*

We are aware that the comparisons we made only provide a limited picture of the differences in the meteorological input data. However, as this is not a study purely on data comparison, we decided to include one measure per variable. For temperature, it is not possible to calculate an annual sum that can be compared. Therefore, the mean daily

difference (which is the same as when the annual mean temperature is calculated first, and the mean difference is calculated then) is given in that case.

*Figure 4: What are the reasons for the different model performance among the countries? What are the governing factors? Simply stating the KGE is higher here or lower there without providing further reasons sounds not helpful.*

We thank the reviewer for their comments. However, note that in the section "Results" we only describe the results of the study, without interpretation. The reasons for the higher or lower model performances – as far as they could be identified – are given in the section "Discussion". Motivated by a comment of Reviewer 2, we are also avoiding the direct comparison of the model performances (i.e., the KGE values) for different regions between each other.

*Figure 6: The important thing is not the exact number of catchments in a country where E-OBS dataset is better or worse than the CAMELS datasets, but why E-OBS is better/worse than the CAMELS in these catchments?*

We thank the reviewer for pointing this out. We think that it can be helpful to know which meteorological input data lead to a more successful streamflow simulation, since the model performance can be interpreted as an aggregated measure for hydrological efficacy (and thus gives an indication of which data may be of a higher quality). Regarding the reasons for the lower or higher model performances, these are discussed in the section "Discussion", where the modelling results are interpreted.

*L275-278: Why is the model performance lower in Great Britain which shows opposite behavior? Please explain.*

In scenario III, we use the (higher) potential evapotranspiration data from EStreams, while in scenario II, we use the (lower) potential evapotranspiration data from CAMELS (in this case, CAMELS-GB). In section 4.4, we explain why for the karstic catchments in Great Britain, the higher potential evapotranspiration data were beneficial. We have added a sentence at L457 and -onwards indicating that for these catchments, potential evapotranspiration was more important than elsewhere to make the link to the explanation provided later.

*Figure 7: Simply stating the station density plays the key role seems not convincing, as the author stated that other factors may also play a role. It would be more interesting to analyze other factors as well? Are the relationships between the station numbers and the KGE statistically significant?*

We have computed the correlations between KGE and catchment descriptors, and this statement is based on that. Specifically, only the correlation with the number of precipitation (and temperature) stations and aridity showed independent and interesting results.

We have added the table with the correlations in the appendix. We have also improved the discussion and also made some improvements in the figure, such as including the Spearman rank coefficient (relationships) and $p$-values (significance) for each subplot (country).

*Figure 8: What about a trend assessment on the data? Is there a significant relationship between model performance and aridity index?*

We have computed the correlation between the two variables (Spearman rank coefficient and $p$-value) and consequently plotted the LOWESS (locally weighted) smooth line for the trend assessment for each subplot in Figure 8.

*L366-369: it is too assertive and not supported by evidence. It is a very simple method to calculate the potential evapotranspiration which does not consider solar radiation impact. It is also too assertive to say different calculation approaches of potential evapotranspiration will not change the results.*

Note that the evidence that the Hargreaves equation is reliable (e.g., in Central Europe) does not origin from the current study, but from the study by Pimentel et al. (2023), cited in this sentence. Furthermore, the choice for the Hargreaves equation was not made for this study, but when the EStreams dataset was published. The statement that the different potential evapotranspiration data did not affect the model performance results strongly is supported by Figure A6.

We have made changes in Section 4.4, including the new references.